# Cryo-EM structure of an active bacterial TIR–STING filament complex

Benjamin R. Morehouse[1,2], Matthew C. J. Yip[3], Alexander F. A. Keszei[3], Nora K. McNamara-Bordewick[2], Sichen Shao[3 ✉] & Philip J. Kranzusch[1,2,4 ✉]

Stimulator of interferon genes (STING) is an antiviral signalling protein that is broadly conserved in both innate immunity in animals and phage defence in prokaryotes[1–4]. Activation of STING requires its assembly into an oligomeric filament structure through binding of a cyclic dinucleotide[4–13], but the molecular basis of STING filament assembly and extension remains unknown. Here we use cryogenic electron microscopy to determine the structure of the active Toll/interleukin-1 receptor (TIR)–STING filament complex from a *Sphingobacterium faecium* cyclic-oligonucleotide-based antiphage signalling system (CBASS) defence operon. Bacterial TIR–STING filament formation is driven by STING interfaces that become exposed on high-affinity recognition of the cognate cyclic dinucleotide signal c-di-GMP. Repeating dimeric STING units stack laterally head-to-head through surface interfaces, which are also essential for human STING tetramer formation and downstream immune signalling in mammals[5]. The active bacterial TIR–STING structure reveals further cross-filament contacts that brace the assembly and coordinate packing of the associated TIR NADase effector domains at the base of the filament to drive $NAD^+$ hydrolysis. STING interface and cross-filament contacts are essential for cell growth arrest in vivo and reveal a stepwise mechanism of activation whereby STING filament assembly is required for subsequent effector activation. Our results define the structural basis of STING filament formation in prokaryotic antiviral signalling.

Activation of STING signalling results in assembly of oligomeric filament structures that have been observed in both human innate immunity and bacterial antiphage defence[4–13]. Key early findings supporting STING oligomerization as a required step of activation include observation of STING puncta formation in cells[9], electrophoresis analysis of STING multimeric complexes[10,11] and artificial activation of STING on fusion to multimerization domains[12]. More recently, insight into the structural basis of mammalian STING oligomerization has been obtained through analysis of STING crystal packing[7] and cryogenic electron microscopy (cryo-EM) structures of tetrameric STING complexes[5,13]. Strict conservation of filament formation in prokaryotic STING signalling suggests that prokaryotic and metazoan STING signalling domains share an ancient mechanism of signal induction[4]. To define the molecular basis of STING filament formation, we reconstituted signalling of the *S. faecium* TIR–STING (*Sf*STING) antiphage effector in vitro and used single-particle cryo-EM to determine the structure of the activated complex (Fig. 1a and Extended Data Figs. 1, 2, 3 and 8). In response to the nucleotide second messenger c-di-GMP produced during cyclic oligonucleotide-based antiphage signalling system (CBASS) immunity, *Sf*STING rapidly assembles into oligomers that form single filaments and antiparallel double-filament structures that make supra-molecular contacts between STING and TIR domains of opposing filaments (Extended Data Figs. 1 and 2). The TIR domains

are not as well resolved in the main double-filament class, probably owing to conformational heterogeneity, and we therefore focused structural analysis on the single-fibre filaments. A 3.3-Å-resolution cryo-EM reconstruction of the dominant class of single-fibre filaments reveals that *Sf*STING oligomerizes through formation of a repeating laterally translated array of parallel stacked protein dimers that buries more than 3,000 Å[2] of surface area between two pairs of dimers and locks the STING cyclic dinucleotide (CDN)-binding domain and associated TIR effector domains into filamentous assemblies capable of reaching greater than 300 nm in length (Fig. 1a).

## Architecture of *Sf*STING filaments

To explain how CDN signal recognition drives filament formation, we determined the cryo-EM structure of *Sf*STING bound to the weakly activating ligand 3′,3′-cGAMP (ref. [4]; Extended Data Figs. 2 and 3). Compared with the modelled open apo state and partially closed 3′,3′-cGAMP-bound conformations, recognition of the signal from the correct nucleotide, c-di-GMP, induces an inward rotation of the *Sf*STING β-strand 'lid' region of about 25° and about 9° respectively and results in formation of a tightly closed complex (Fig. 1b and Extended Data Fig. 4). Tighter lid closure is driven by repositioning of the highly conserved *Sf*STING lid domain residue R234 to form base-specific

[1]Department of Microbiology, Harvard Medical School, Boston, MA, USA. [2]Department of Cancer Immunology and Virology, Dana-Farber Cancer Institute, Boston, MA, USA. [3]Department of Cell Biology, Harvard Medical School, Boston, MA, USA. [4]Parker Institute for Cancer Immunotherapy at Dana-Farber Cancer Institute, Boston, MA, USA. ✉e-mail: sichen_shao@hms.harvard.edu; philip_kranzusch@dfci.harvard.edu

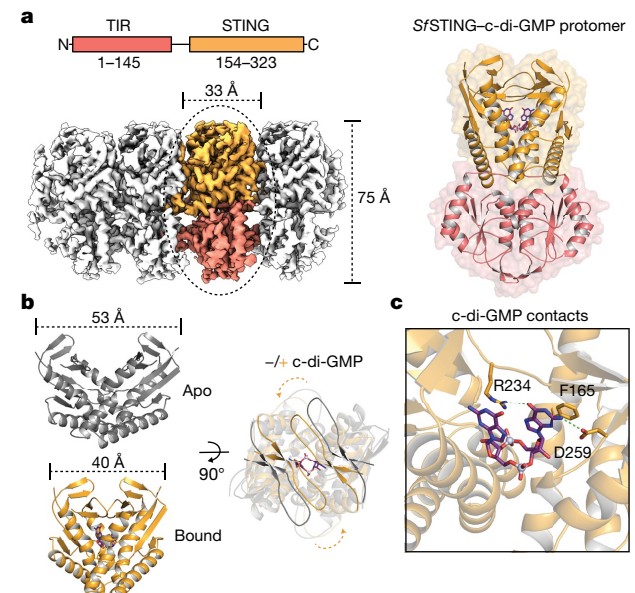

**Fig. 1 | Cryo-EM structure of the active TIR–STING filament. a**, Left, *Sf*STING domain organization (top) and cryo-EM density map of the active *Sf*STING–c-di-GMP filament complex (bottom). The colouring of the density for one dimer is used to highlight the filament organization, with TIR in pink and the STING CBD in orange. Right, an isolated *Sf*STING protomer (dimer) rotated 90° along the vertical axis relative to the orientation of the filament on the left. c-di-GMP is shown as a purple stick model. **b**, Left, a comparison of the apo (grey; top) versus the c-di-GMP-bound (orange, inside grey; bottom) *Sf*STING CBD highlighting the V-shaped homodimer closing in on the ligand. The apo *Sf*STING CBD was modelled through structural alignment with the crystal structure of a related prokaryotic TIR–STING from *C. granulosa* (Protein Data Bank (PDB) 6WT4)[4]. Right, a top-down view highlighting the closure of the β-strand 'lid' (90° rotated). **c**, A close-up view of the c-di-GMP-binding pocket of *Sf*STING. Several side chains make direct contacts to the c-di-GMP. Symmetry-related contacts are not shown for clarity.

contacts with the guanosine Hoogsteen edge, mirroring interactions required for high-affinity complex formation between human STING and the cGAS product 2′,3′-cGAMP (refs. [14,15]; Fig. 1c). The partially closed 3′,3′-cGAMP-bound *Sf*STING conformation allows incomplete oligomerization and is incompatible with stable filamentous packing beyond about 4–6 units (or about 20 nm). Compared to the fully active *Sf*STING–c-di-GMP filament structure, 3′,3′-cGAMP recognition induces partial closure of the lid domain and an overall conformation of the STING CDN-binding domain (CBD) that probably weakens contact sites observed for c-di-GMP-induced filaments. Additionally, a lack of well-defined density for the TIR domains in the 3′,3′-cGAMP filaments suggests conformational flexibility that may impact stability of the filaments. *Sf*STING binds to c-di-GMP with about 300 nM apparent affinity ($K_d$) and a low nanomolar concentration of c-di-GMP is sufficient to initiate robust NADase activity[4]. Our previous results demonstrate that 3′,3′-cGAMP binds with slightly weaker affinity (about 700 nM $K_d$) and is unable to induce robust TIR NADase activity, findings that are now explained by our cryo-EM analysis. Thus, complete closure of the lid domain and structural compression around the high-affinity ligand c-di-GMP is essential to translate CDN recognition into a conformation sufficient to seed STING protein-filament formation and downstream signal induction.

In the *Sf*STING–c-di-GMP complex, individual dimeric units adopt a two-fold symmetric conformation and form the basic repeating building block of the filament structure (Fig. 1a, right). In each *Sf*STING dimer unit, the canonical V-shaped STING CBD is positioned above two TIR enzymatic NADase domains that dock against the base of the

receptor (Fig. 1a and Extended Data Figs. 4 and 5). The *Sf*STING CBD contains a unique β-hairpin insertion immediately following the stem dimerization helix, but otherwise adopts the same minimized fold and highly conserved CDN-binding pocket previously observed in crystal structures of *Flavobacteriaceae* sp. and *Capnocytophaga granulosa* bacterial STING (ref. [4]). In the active-state *Sf*STING filament structure, a short linker sequence connects the α-helix stem of each STING domain to the TIR effector domain (Extended Data Fig. 4e). Previously, structures of a TIR–STING homologue from the oyster *Crassostrea gigas* and the human transmembrane domain-containing STING in inactive states revealed a twisted linker sequence that connects the effector domain to the STING domain located across the dimeric interface[4,5,13] (Extended Data Fig. 4e). The active-state conformation of the chicken *Gallus gallus* STING–2′,3′-cGAMP complex exhibits a parallel linker orientation similar to the *Sf*STING filament, suggesting that parallel linker orientation is a defining feature of both prokaryotic and metazoan STING activation[5,6].

## Mechanism of *Sf*STING oligomerization

TIRs are widespread NADase effector domains encoded in CBASS, Pycsar and Thoeris antiphage defence systems[4,16,17], but no previous TIR active-state structures exist to explain the mechanism of NAD+ hydrolysis. The *Sf*STING TIR domain is most closely related to plant immune proteins and secreted bacterial effectors that catalyse glycosidic bond hydrolysis during immune defence and interspecies conflict[17–21]. The *Sf*STING residues F83, F85, L87 and L89 within the highly conserved TIR helix αC interface form extensive hydrophobic packing interactions that bridge the dimeric unit, and the *Sf*STING TIR domain also contains a unique βD′ and βE′ strand insertion in the TIR 'CC loop' that further extends the dimer interface (Fig. 2a–c and Extended Data Fig. 5). Structural comparison with the human SARM1 TIR–ribose structure demonstrates that the *Sf*STING NAD+-binding pocket is formed by two regions: a set of hydrophobic residues, F6, W33, F37 and L47, positioned to stack the substrate nicotinamide; and a set of hydrophilic residues, S10, R78 and N80, positioned to coordinate the phosphodiester linkage and ribose of the adenosine base[19] (Fig. 2d). In addition to the highly conserved *Sf*STING catalytic residue E84, the NADase active site is completed by residue D110 from the opposite TIR dimer mate (Fig. 2d and Extended Data Fig. 5).

The active *Sf*STING–c-di-GMP structure reveals a series of protein–protein interfaces that explain a shared mechanism of STING filament formation. The primary STING filament interface occurs along two surfaces that pack between adjacent *Sf*STING dimeric units and drive lateral head-to-head oligomerization (Fig. 3a and Extended Data Fig. 6). These surfaces centred around the hydrophobic residues V280 and A309 are exposed in the closed *Sf*STING domain conformation, explaining a key mechanism that couples c-di-GMP recognition and filament nucleation (Extended Data Fig. 6). Individual STING-domain protomers (STING$_a$ and STING$_b$) within the *Sf*STING filament are also bridged by an electrostatic interaction between STING$_a$ R307 and STING$_b$ E290 (Fig. 3b). Notably, the previous cryo-EM structure of a chicken STING–2′,3′-cGAMP tetramer contains residues Q278 and D280 involved in a similar interaction and hydrophobic surfaces packed along the same STING–STING protein interface, revealing remarkable conservation of an ancient mechanism of STING oligomerization[5] (Fig. 3a,b). In the full *Sf*STING filament structure, the STING domains are more tightly packed compared to those in the minimal chicken/human STING tetramer model. Additionally, a modest approximately 2° shift between packed *Sf*STING dimeric units is observed in both the single-fibre and wrapped double-fibre cryo-EM reconstructions, resulting in the active *Sf*STING filament structure adopting a slight curve (Extended Data Fig. 1). Assembly of the complete *Sf*STING filament allows formation of a second cross-filament interface between the STING domain residues N278 and Q279 and residue E95 in the TIR domain associated with the adjacent protomer (Fig. 3c and Extended Data Fig. 6). Cross-filament

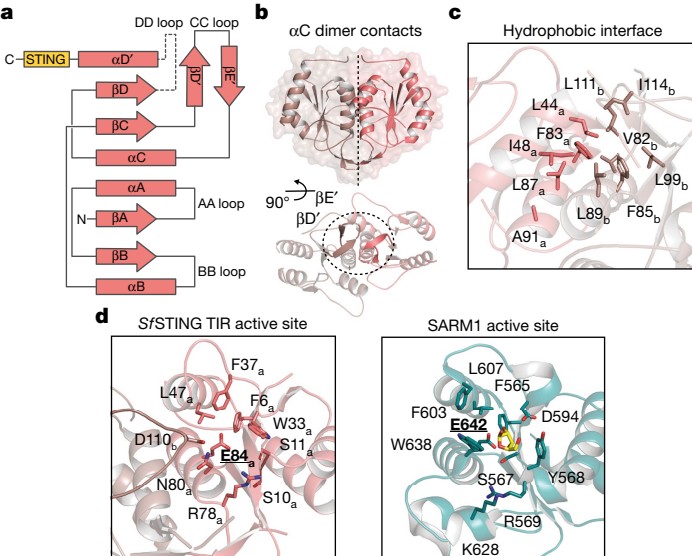

**Fig. 2 | Sf STING TIR NADase active-site architecture. a**, A topology diagram of the secondary structure of the Sf STING TIR domain. The DD loop is shown as a dotted line to indicate the lack of observed density/unbuilt portion of the structure. β-strands are shown as arrows, and α-helices are shown as rectangles. **b**, A global view of the TIR intradimer contact interface with a rotated view highlighting the unique CC loop structure. Each monomer is separately coloured for clarity. **c**, The core dimeric interface formed by αC is lined with nonpolar residues. **d**, Comparison of the Sf STING TIR NAD[+]-binding pocket to human SARM1 (PDB 6O0Q). Catalytic glutamate residues are in bold and underlined. Sf STING D110_b of the opposing monomer projects inwards to complete the binding pocket. The SARM1 structure shows a ribose molecule (yellow) indicating the likely binding position for the ribose and nicotinamide base of NAD[+].

TIR_a–TIR_b interactions are also formed between two flexible Sf STING TIR loops (BB loop: P32_a–G43_a; and DD loop: A101_b–K118_b) that stack on top of one another (Fig. 3d). Comparison of the active Sf STING–c-di-GMP complex with the inactive apo C. granulosa bacterial STING structure reveals substantial rearrangements in the STING domain necessary to enable reorganization and TIR-domain packing[4] (Extended Data Fig. 4). Close packing is required to allow the TIR D110 loop to reach across and complete the dimer-mate active site, providing an explanation for how Sf STING filament formation triggers NADase domain activation (Figs. 2d and 3d and Extended Data Fig. 5).

## Filamentation controls NADase activity

We next combined the bacterial STING filament structure with biochemical and cellular analysis of Sf STING function to establish a molecular model of STING activation. Measuring degradation of a fluorescent NAD[+] analogue, we observed that Sf STING alterations predicted to disrupt STING–STING, STING–TIR and TIR–TIR cross-filament interaction surfaces each strongly inhibit NADase enzymatic activity in vitro (Fig. 4a and Extended Data Fig. 7). The Sf STING substitutions V280D, E290K and R307E within the STING oligomerization interface disrupted all detectable NAD[+] hydrolysis. Likewise, Sf STING variants with substitutions in the STING–STING interface (N208D and A309R) and STING–TIR interface (R52E, K142D, N278E, Q279E and D285K) exhibit weak NADase activity only at 10–100× protein concentration, suggesting defects in the ability to oligomerize and catalyse NAD[+] cleavage (Fig. 4a and Extended Data Fig. 7). Each Sf STING filament interface mutant retains the ability to form a stable, high-affinity complex with c-di-GMP, demonstrating that inhibition of NADase function is not due to impaired protein stability or ligand interaction (Extended Data Fig. 7 and Supplementary Fig. 1). Negative-stain electron microscopy analysis confirmed that the absence of NADase activity is a direct result of Sf STING interface mutants specifically losing the ability to form an active filament complex (Fig. 4b and Extended Data Fig. 7). Replacement of the TIR BB loop within the principal TIR_a–TIR_b interaction site with a glycine linker sequence (ΔA36–K41) resulted in complete disruption of NADase function (Fig. 4a and Extended Data Fig. 7). However, this Sf STING TIR mutant retains the ability to oligomerize into a filament in the presence of c-di-GMP, demonstrating that STING–STING interactions are the main driver of filamentation and that secondary TIR–TIR cross-filament interactions are required only for induction

**Fig. 3 | Bacterial STING and human STING share an ancient mechanism of filament formation. a**, The tetramer interfaces formed between the filaments within the CBD are similar for Sf STING and human STING. The 'tetramer' modelled here for human STING (blue, PDB 6NT5) is based on cryo-EM observation for chicken STING (PDB 6NT8). The dashed squares indicate cross-filament contact surfaces. **b**–**d**, Close-up views of Sf STING cross-filament interfaces including electrostatic contacts coordinating STING-to-STING (**b**), STING-to-TIR (**c**) and TIR-to-TIR (**d**) interactions. In TIR-to-TIR contacts, the BB loop and the DD loop of opposing TIR monomers reside flush against each other with only one direct contact from T115 on the DD loop to the backbone amide bond of N40 on the BB loop. Schematic depictions of cross-filament domain contacts (indicated by arrows) are shown in the upper-left insets of **b**–**d**.

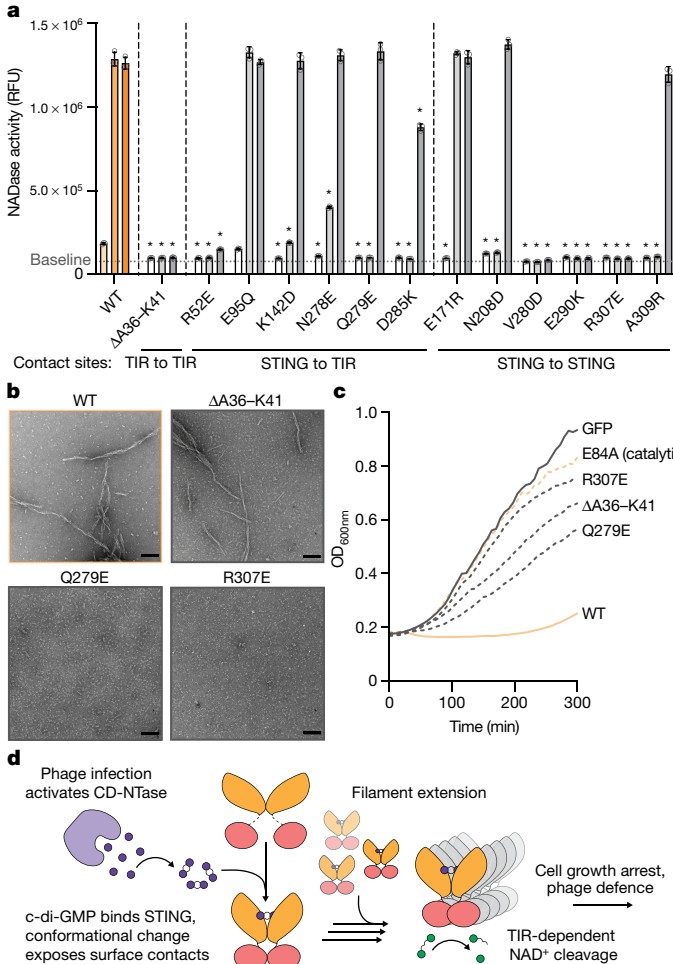

**Fig. 4 | TIR-mediated NAD⁺ cleavage is driven by STING oligomerization.**
**a**, A bar-graph representation of NADase activity measured with the NAD$^+$ fluorescent analogue nicotinamide 1,$N^6$-ethenoadenine dinucleotide for a panel of *Sf*STING residue substitutions at a range of enzyme concentrations. NADase activity is measured as fluorescence intensity (relative fluorescence units (RFU)) at about 5 min. Each bar within a set corresponds to 0.1, 1 or 10 μM enzyme. The baseline threshold indicates the background fluorescent signal. The error bars indicate the standard deviation for the average of three biological replicates each with three technical replicates. *$P < 0.0001$ (one-way analysis of variance comparing the mean value for each mutant to that of the wild type (WT) at the same protein concentration); $P$ values for bars without an asterisk are greater than 0.05 and considered not significantly different.
**b**, Negative-stain micrograph images for select *Sf*STING mutants and the wild type in the presence of c-di-GMP. Scale bars, 100 nm. Each image is representative of $n = 6$ micrograph images. **c**, Cell growth curves for *E. coli* cultures expressing a select panel of *Sf*STING mutants and the wild type. The data are representative of more than three independent biological replicates each with three technical replicates. The mean curve is shown for one biological replicate. **d**, A schematic model describing the process of *Sf*STING NADase activation through ligand binding and filament formation. CD-NTase, cGAS/DncV-like nucleotidyltransferase.

of NADase catalysis. We also observe that a D110A mutant retains the ability to recognize c-di-GMP and form long protein filaments but loses all ability to initiate NADase activity, providing further evidence for the essential role of filament formation in the activation mechanism of *Sf*STING (Extended Data Fig. 7). We expressed *Sf*STING in *Escherichia coli*, a bacterium that constitutively produces the activating ligand c-di-GMP, and confirmed that each *Sf*STING filament interaction interface is essential for STING-induced growth arrest in vivo (Fig. 4c and Extended Data Fig. 7).

Our results provide a complete structural model of bacterial STING filament formation and effector domain activation in CBASS antiphage defence (Fig. 4d). STING-mediated antiphage defence begins when the associated CBASS protein CdnE, a cGAS/DncV-like nucleotidyltransferase that recognizes a yet unknown phage cue, senses bacteriophage infection and initiates synthesis of the antiviral nucleotide second messenger c-di-GMP (refs. [4,22,23]). c-di-GMP is a high-affinity ligand that binds STING in a central chamber formed at the receptor homodimeric interface. Cognate CDN signal recognition induces a conformational change in the STING β-strand lid domain that envelopes c-di-GMP in a closed receptor complex. Next, the conformational change induced on complete lid closure exposes surface contact sites that create an interface for nucleating STING filament formation. STING filament extension is driven primarily by STING–STING contacts and cross-filament contacts between STING and the associated TIR effector domain. Finally, filament assembly leads to TIR–TIR interactions that rearrange the NADase active site to stimulate NAD⁺ degradation and an abortive infection response that prevents phage propagation. In further support of our model of TIR NADase activation in CBASS antiphage defence, another study has determined the high-resolution structure of a distinct CBASS effector named TIR-SAVED that demonstrates that cyclic oligonucleotide binding induces a curved protein filament responsible for TIR NADase activation[24]. In animal cells, protein oligomerization has emerged as a general principle controlling rapid induction of innate immune signalling[25]. Our structural analysis of bacterial STING activation defines the molecular basis of STING filament formation and demonstrates remarkable conservation of oligomerization as a unifying mechanism controlling both prokaryotic and metazoan antiviral immune defence.

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

## Methods

### Synthetic nucleotide ligands

Synthetic CDN ligands were purchased from Biolog Life Science Institute: c-di-GMP (catalogue number C 057) and 3′,3′-cGAMP (catalogue number C 117). Benzamide adenine dinucleotide was a gift from Frank Schwede (Biolog Life Science Institute).

### Protein expression and purification

Recombinant bacterial *Sf*STING protein was recombinantly expressed and purified as previously described[4]. Briefly, all constructs were cloned using Gibson assembly into a modified pET16 vector for expression of recombinant amino-terminal 6×His-fusion proteins in BL21-CodonPlus(DE3)-RIL *E. coli* (Agilent)[26]. The TIR-to-TIR cross-filament contact mutant ΔA36–K41 was designed as a glycine-serine loop replacement (D35-GSGG-S42). Inoculated 1-l M9ZB cultures (0.5% glycerol, 1% Cas-amino acids, 47.8 mM $Na_2HPO_4$, 22 mM $KH_2PO_4$, 18.7 mM $NH_4Cl$, 85.6 mM NaCl, 2 mM $MgSO_4$ and trace metals, supplemented with 30 mM nicotinamide to limit TIR toxicity) were grown at 37 °C with 230 r.p.m. shaking. Cultures reaching an optical density at 600 nm ($OD_{600nm}$) > 2.5 were induced with a final IPTG concentration of 500 μM and incubated at 16 °C overnight at 230 r.p.m. Collected bacterial pellets were sonicated in lysis buffer (20 mM HEPES-KOH pH 7.5, 400 mM NaCl, 30 mM imidazole, 10% glycerol and 1 mM dithiothreitol) and purified by gravity flow over Ni-NTA resin (Qiagen). Resin was washed once with lysis buffer supplemented to 1 M NaCl, and recombinant protein was eluted with 300 mM imidazole. Protein was dialysed overnight at 4 °C (20 mM HEPES-KOH pH 7.5, 250 mM KCl, 10% glycerol and 1 mM dithiothreitol). Dialysed protein was concentrated with 30-kDa-cutoff Amicon centrifuge filters (Millipore) before loading onto a 16/600 Superdex 200 size-exclusion column (Cytiva) equilibrated in gel filtration buffer (20 mM HEPES-KOH pH 7.5, 250 mM KCl, 1 mM TCEP). Protein purity was assessed by denaturing gel before concentrating samples to >10 mg ml$^{-1}$ and flash freezing in liquid nitrogen for storage at −80 °C.

### Cryo-EM sample preparation and data collection

On exposure to the activating ligand c-di-GMP, solutions of purified *Sf*STING immediately begin filament formation and become visibly cloudy. For the first c-di-GMP dataset, *Sf*STING at 1 mg ml$^{-1}$ was rapidly mixed with a 3× molar concentration of c-di-GMP (84 μM), immediately applied to glow-discharged 1.2/1.3 Cu 300 mesh grids (Quantifoil), and frozen in liquid ethane within 10 s of mixing using a Vitrobot Mark IV (Thermo Fisher) set at 4 °C and 100% humidity with no wait time, 3 s blot time and +8 blot force. For the second c-di-GMP dataset, *Sf*STING at 1 mg ml$^{-1}$ was pre-incubated with 1 mM benzamide adenine dinucleotide before rapid mixing with 84 μM c-di-GMP and frozen as above. Semi-automated data collection was performed with SerialEM v3.8.5 and v3.8.6. Grids were imaged on a Titan Krios (Thermo Fisher) operating at 300 kV equipped with a BioQuantum K3 imaging filter with a 20-eV slit width and a K3 summit direct electron detector (Gatan) in counting mode at a nominal magnification of 105,000× corresponding to a calibrated pixel size of 0.825 Å. For the first dataset, a total exposure time of 1.6 s, corresponding to a total dose of 55.5 electrons Å$^{-2}$, was fractionated over 49 frames. For the second dataset, a total exposure time of 1.29 s, corresponding to 51.7 electrons Å$^{-2}$, was fractionated over 51 frames. The defocus targets were −1.2 to −2.1 μm for the first dataset and −1.2 to −2.5 μm for the second dataset.

For the 3′,3′-cGAMP dataset, *Sf*STING at 1 mg ml$^{-1}$ was rapidly mixed with a 3× molar concentration of 3′,3′-cGAMP (84 μM) and frozen as described above. The 3′,3′-cGAMP dataset was collected on a Talos Arctica (Thermo Fisher) operating at 200 kV equipped with a K3 direct electron detector (Gatan) in counting mode at a nominal magnification of 36,000× corresponding to a calibrated pixel size of 1.1 Å. A total

exposure time of 4.494 s, corresponding to a total dose of 52.9 electrons Å$^{-2}$, was fractionated into 50 frames. The defocus targets were −1.4 to −2.6 μm.

### Cryo-EM image processing and model building

Data processing was performed in cryoSPARC v3.1.018 (ref. [27]) and RELION-3.1 (ref. [28]). For the c-di-GMP datasets, patch-based motion correction and CTF estimation was performed in cryoSPARC. Micrographs with severe contamination or poor contrast transfer function (CTF) fits were removed. Automated particle picking was performed in cryoSPARC with the template picker, using templates generated from either the filament tracer (first dataset) or the blob-based picker (second dataset). The particles were extracted with a box size of 320 and downsampled to a box size of 160 for initial two-dimensional (2D) classification and refinement steps.

For the c-di-GMP-bound *Sf*STING single-filament reconstruction, particle coordinates from the filament tracer and template-based picking in the first c-di-GMP dataset were combined, and duplicate coordinates closer than 40 Å were removed. A total of 277,287 coordinates corresponding to single-filament classes after heterogeneous refinement were imported into RELION. The second dataset had more bundled filaments and did not contribute to the single-filament reconstruction. Global and local (12 × 8 patches) motion correction was repeated in RELION using MotionCor2 v1.4.0 (ref. [29]), followed by CTF estimation with GCTF v1.06 (ref. [30]). After 2D classification and 3D refinement, 270,695 particles were subjected to signal subtraction using a mask around the central filament, followed by 3D classification without alignment. A total of 206,965 particles were reverted and subjected to two rounds of CTF refinement and a round of Bayesian polishing. One 3D classification without alignments was performed with the polished particles using a mask around the central filament. A class containing 26,447 particles that best resolved both the TIR and STING domains was selected for a final round of 3D refinement. In our analysis, *Sf*STING activation is observed as individual filaments that range in size with some filaments reaching >300 nm in length (about 85 dimer copies, about 6.3 MDa). Particles selected for processing and high-resolution structural analysis include density for at least 5 *Sf*STING dimer copies.

For the c-di-GMP-bound *Sf*STING double-filament reconstruction, multiple rounds of heterogeneous refinement were performed independently on each dataset in cryoSPARC to isolate particles contributing to the best reconstructions of a double filament after 3D non-uniform refinement. The final reconstructions contained 176,549 particles in the first dataset and 178,579 particles in the second dataset. As no further density corresponding to the benzamide adenine dinucleotide analogue or other differences were observed in the maps, the double-filament particle coordinates from the two datasets were combined and subjected to local motion correction, CTF refinement and non-uniform refinement. For all datasets, attempts to apply symmetry or helical parameters resulted in inferior reconstructions because the *Sf*STING dimers are not exactly symmetrically related in the oligomeric complexes.

For the 3′,3′-cGAMP dataset, patch-based motion correction and CTF estimation was performed in cryoSPARC. Micrographs with severe contamination or poor CTF fits were removed. Automated particle picking was performed in cryoSPARC with the template picker using templates generated from the blob-based picker. The particles were extracted with a box size of 280 and subjected to 2D classification followed by ab initio reconstruction and 3D non-uniform refinement. The resulting map and corresponding 261,685 particle coordinates were exported to RELION. Global and local (12 × 8 patches) motion correction and CTF estimation was repeated in RELION using MotionCor2 and GCTF respectively. After a round of 3D classification, 105,567 particles in classes with clear density for all four strands were subjected to CTF refinement, Bayesian

polishing and 3D refinement without and with a mask around the two most defined strands.

The *Fs*STING (PDB 6WT5) CBD was used as a starting model docked into the single-fibre c-di-GMP-bound *Sf*STING density in Coot followed by iterative manual model building[31]. The c-di-GMP-bound *Sf*STING dimer was used as the starting model in the c-di-GMP-bound double filament and 3′,3′-cGAMP-bound oligomer. In the c-di-GMP double filament, individual secondary structure elements of the TIR domains of the central dimer that interacts with the STING domain of the other filament were placed by rigid fitting and manually adjusted in Coot. N-terminal portions of the TIR domain where side chains were not visible were converted to polyalanine. The TIR domains of all other *Sf*STING dimers in the c-di-GMP double filament and 3′,3′-cGAMP oligomer were removed. The 3′,3′-cGAMP-bound *Sf*STING dimers probably contain a combination of the 3′,3′-cGAMP orientation modelled and an approximately 180° rotation. Multiple rounds of Phenix real-space refine[32] was applied with manual correction in Coot in between. Model validation was performed in Phenix using MolProbity (ref. [33]). Figure panels were generated using ChimeraX (ref. [34]) and PYMOL (v2.5.1). Software for data processing and modelling was configured in part by SBGrid (ref. [35]).

### Analysis of TIR NAD$^+$ cleavage activity with fluorescent nicotinamide 1,$N^6$-ethenoadenine dinucleotide

Plate reader reactions to assess NADase function were prepared as described previously[4]. Reactions were built in 50 µl final volume with reaction buffer (20 mM HEPES-KOH pH 7.5, 100 mM KCl), 500 µM nicotinamide 1,$N^6$-ethenoadenine dinucleotide; (ε-NAD, Sigma), 0.1–10 µM enzyme and 20 µM c-di-GMP. Reactions were prepared as master mixes in PCR-tube strips and initiated by adding nicotinamide 1,$N^6$-ethenoadenine dinucleotide immediately before placing into the plate reader. Fluorescence emission at 410 nm was read continuously over 40 min using a Synergy H1 Hybrid Multi-Mode Reader (BioTek) after excitation at 300 nm. Plots were generated with GraphPad Prism 9.3.0.

### Electrophoretic mobility shift assay

*Sf*STING interactions with radiolabelled c-di-GMP were monitored by electrophoretic mobility shift assay as previously described[4]. In brief, 10-µl reactions contained 1× buffer (5 mM Mg(OAc)$_2$, 50 mM Tris-HCl pH 7.5, 50 mM KCl) with a final protein concentration of 20 µM and about 1 µM α$^{32}$-P-labelled c-di-GMP generated by overnight reaction of purified *Vibrio cholerae* DncV with GTP (about 0.1 µCi). Reactions were incubated for 5 min at 25 °C and separated on a 6% nondenaturing polyacrylamide gel held at 100 V for 45 min in 0.5× TBE buffer. Gels were fixed (40% ethanol and 10% glacial acetic acid) before drying at 80 °C for 1 h. Dried gels were then exposed to a phosphor storage screen and imaged on a Typhoon Trio Variable Mode Imager (GE Healthcare).

### Negative-stain EM sample preparation, data collection and image analysis

Wild-type or mutant *Sf*STING (1 µM) was incubated with 10 µM c-di-GMP in buffer (20 mM HEPES-KOH pH 7.5, 250 mM KCl, 1 mM TCEP) for 15 min on ice. The mixture was then directly applied to a glow-discharged (30 s, 30 mA) 400-mesh Cu grid (Electron Microscopy Sciences, EMS-400Cu) coated with an approximately 10-nm layer of continuous carbon (Safematic CCU-010) for 30 s. After side blotting, the grid was immediately stained with 1.5% uranyl formate and then blotted again from the side. Staining was repeated twice with a 30-s incubation with uranyl formate in the final staining step. EM images were collected on a FEI Tecnai T12 microscope operating at 120 keV and equipped with a Gatan 4K × 4K CCD camera at a nominal magnification of 52,000× corresponding to a pixel size of 2.13 Å and at a defocus of about 1 µm.

### STING toxicity analysis in *E. coli*

*Sf*STING and mutant constructs as well as an sfGFP negative-control construct were cloned into pET vectors for IPTG-inducible expression. *E. coli* BL21 (DE3) (NEB) were transformed with these plasmids and then plated on LB medium plates supplemented with 100 µg ml$^{-1}$ ampicillin. After overnight incubation, three colonies from these plates were used to inoculate 5-ml MDG liquid cultures (0.5% glucose, 25 mM Na$_2$HPO$_4$, 25 mM KH$_2$PO$_4$, 50 mM NH$_4$Cl, 5 mM Na$_2$SO$_4$, 2 mM MgSO$_4$, 0.25% aspartic acid and trace metals) supplemented with 100 µg ml$^{-1}$ ampicillin and grown overnight at 37 °C with 230 r.p.m. shaking. Cultures were diluted 1:50 into fresh M9ZB medium (supplemented with 100 µg ml$^{-1}$ ampicillin) and grown for 3 h at 37 °C with 230 r.p.m. shaking. Cultures were then diluted to a uniform OD$_{600nm}$ in M9ZB medium and further diluted 1:5 into fresh M9ZB medium supplemented with 5 µM IPTG to induce protein expression. A 200 µl volume of induced culture was added to a 96-well plate in technical triplicate and OD$_{600nm}$ was recorded over 300 min in a Synergy H1 Hybrid Multi-Mode Plate Reader (BioTek) shaking at 37 °C. Plots were generated with GraphPad Prism 9.3.0.

### Reporting summary

Further information on research design is available in the Nature Research Reporting Summary linked to this paper.

### Data availability

Coordinates and density maps have been deposited with the PDB and the Protein Data Bank in Europe under the following accession codes: *Sf*STING single fibres with c-di-GMP—7UN8 and EMD-26616; *Sf*STING double fibres with c-di-GMP—7UN9 and EMD-26617; *Sf*STING short fibres with 3′,3′-cGAMP (masked)—7UNA and EMD-26618; and *Sf*STING short fibres with 3′,3′-cGAMP—EMD-26619. Data that support the findings of this study are available within the article and its Extended Data and Supplementary Information. Source data are provided with this paper.

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

**Acknowledgements** We are grateful to members of the laboratories of S.S and P.J.K. for helpful comments and discussion. Cryo-EM data collection was performed at the Harvard Center for Cryo-EM. Negative-stain images were collected at the Molecular Electron Microscopy Suite at Harvard Medical School. Cryo-EM data processing was supported by SBGrid. The work was funded by grants to P.J.K. from the Pew Biomedical Scholars programme, the Burroughs Wellcome Fund PATH programme, The Mathers Foundation, The Mark Foundation for Cancer Research, the Parker Institute for Cancer Immunotherapy and the National Institutes of Health (1DP2GM146250-01) and grants to S.S. from the Vallee Foundation, the Packard Foundation and the National Institutes of Health (1DP2GM137415). B.R.M. is supported as a Ruth L. Kirschstein NRSA Postdoctoral Fellow (NIH F32GM133063). M.C.J.Y. is supported by an American Heart Association predoctoral fellowship (287375208).

**Author contributions** Experiments were designed and conceived by B.R.M., S.S. and P.J.K. NAD[+] cleavage and c-di-GMP binding assays were performed by B.R.M. Samples for cryo-EM were prepared by M.C.J.Y. and A.F.A.K. with assistance from B.R.M. EM data collection and processing were conducted by M.C.J.Y., A.F.A.K. and S.S. and analysis was conducted by B.R.M. and S.S. STING cellular toxicity analysis was performed by N.K.M.-B. The manuscript was written by B.R.M., S.S. and P.J.K. All authors contributed to editing the manuscript and support the conclusions.

**Competing interests** The authors declare no competing interests.

**Additional information**
**Correspondence and requests for materials** should be addressed to Sichen Shao or Philip J. Kranzusch.

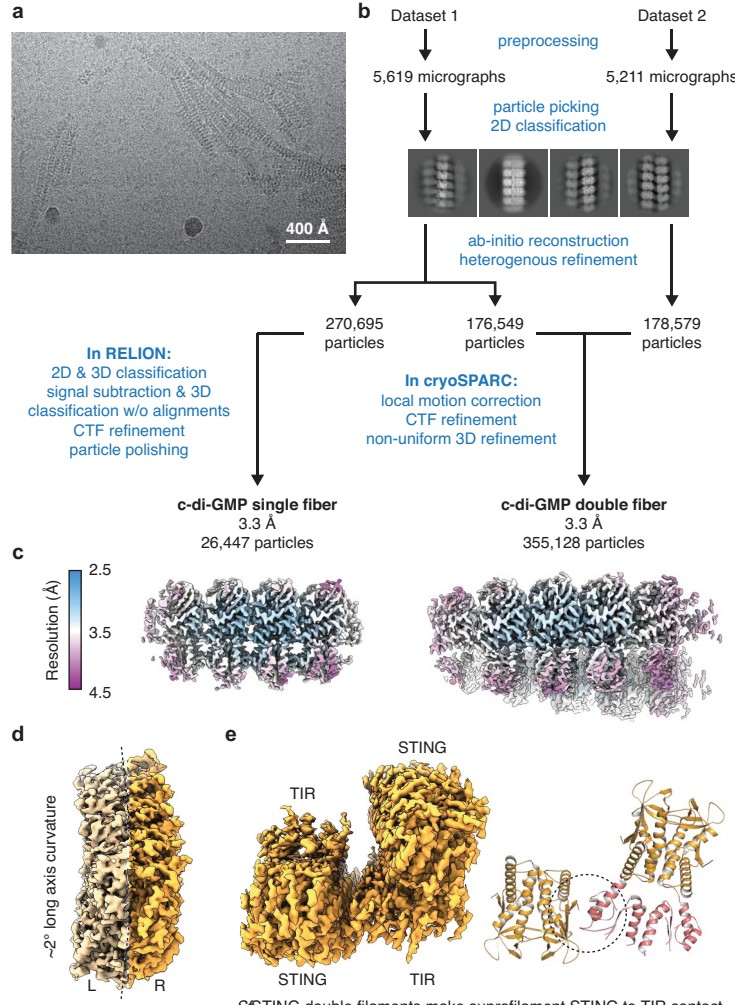

**Extended Data Fig. 1 | Cryo-EM data processing for *Sf*STING with c-di-GMP.**
**a**, Section of a representative electron micrograph (*n* = 10,839 from 2 datasets) of *Sf*STING filaments formed in the presence of c-di-GMP. **b**, Data processing scheme used to generate c). **c**, Reconstructions of single and double *Sf*STING filaments bound to c-di-GMP, coloured by local resolution. **d**, *Sf*STING single fibre curvature is evident along the pseudosymmetry C2 axis. L (left) and R (right) monomers are false coloured for clarity. **e**, *Sf*STING double fibres make additional (limited) STING to TIR contacts. Lack of sufficiently resolved density for the TIR domains limits analysis of this suprafilament interaction.

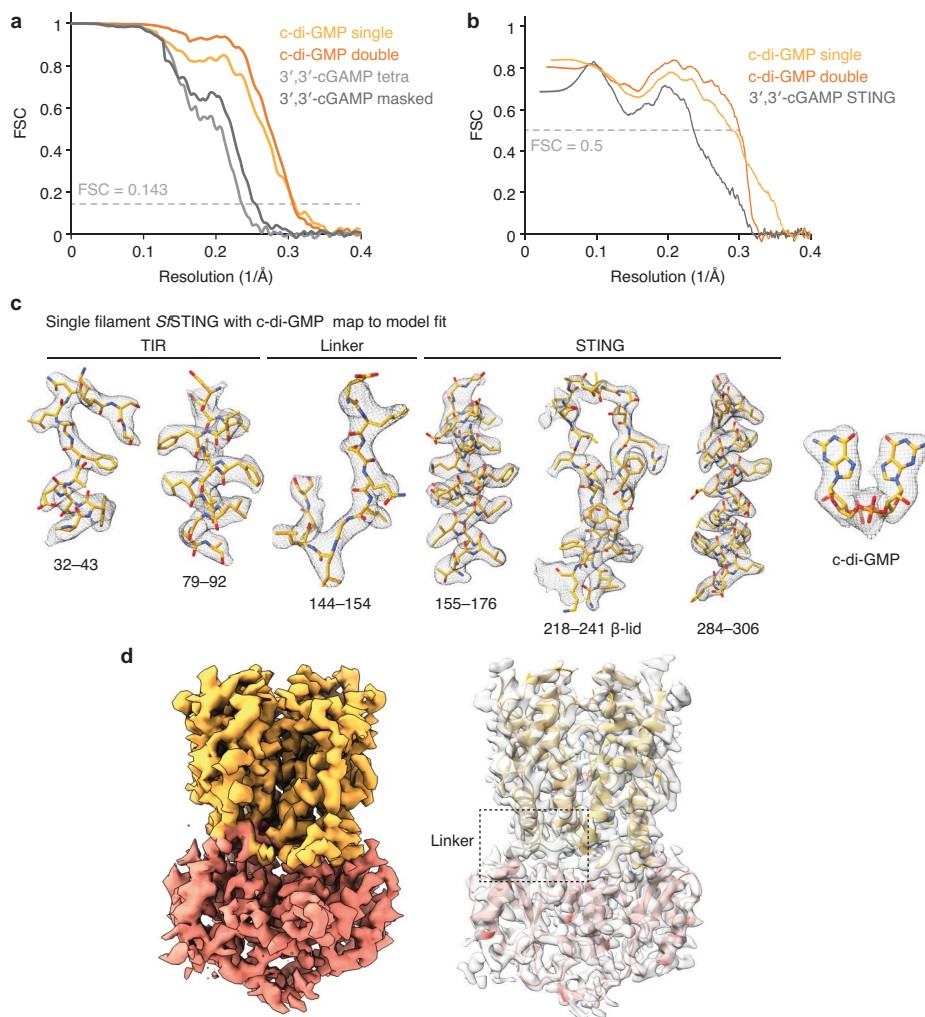

**a**, c-di-GMP single / c-di-GMP double / 3′,3′-cGAMP tetra / 3′,3′-cGAMP masked

FSC = 0.143

**b**, c-di-GMP single / c-di-GMP double / 3′,3′-cGAMP STING

FSC = 0.5

**c**, Single filament *Sf*STING with c-di-GMP map to model fit

TIR | Linker | STING

32–43    79–92    144–154    155–176    218–241 β-lid    284–306    c-di-GMP

**d**, Linker

**Extended Data Fig. 2 | Cryo-EM map quality and model to map fitting.**
**a**, Fourier shell correlation (FSC) curves versus resolution (1/Å) of indicated EM maps. Resolution was estimated at FSC = 0.143. **b**, Model versus map FSC curves. **c**, Examples of map to model fit quality for select regions of the single fibre *Sf*STING complex with c-di-GMP (contoured at 7σ). **d**, Overall density map for filament protomer *Sf*STING dimer. STING CBD in orange and TIR in pink. Transparent density with dimer model (contoured at 7σ, right). Linker region sequence (144–176) highlighted in green and boxed.

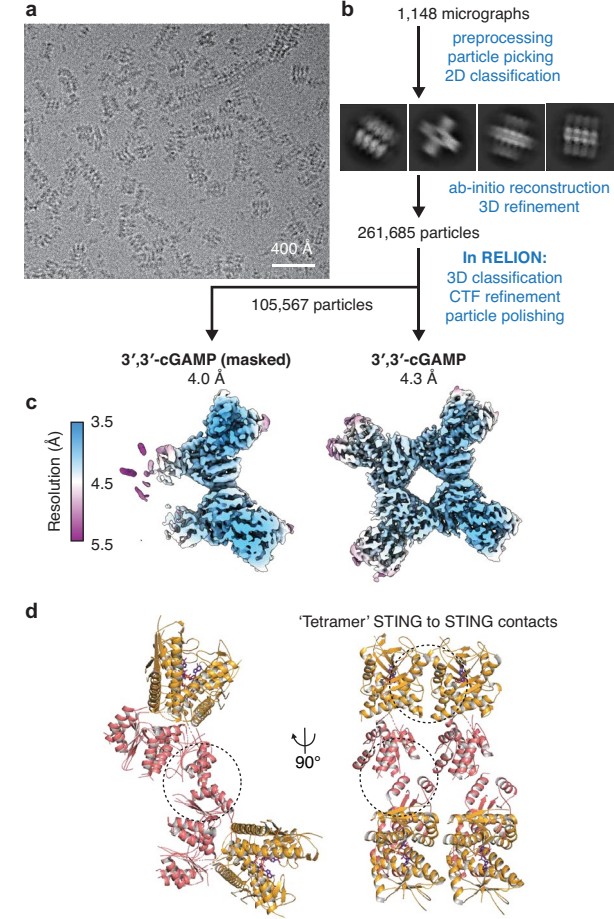

**a**

400 Å

**b**

1,148 micrographs

preprocessing
particle picking
2D classification

ab-initio reconstruction
3D refinement

261,685 particles

**In RELION:**
3D classification
CTF refinement
particle polishing

105,567 particles

3′,3′-cGAMP (masked)
4.0 Å

3′,3′-cGAMP
4.3 Å

**c**

Resolution (Å)

3.5
4.5
5.5

**d**

'Tetramer' STING to STING contacts

90°

TIR to TIR suprafilament contacts not observed with c-di-GMP

**Extended Data Fig. 3 | Cryo-EM data processing for *Sf*STING with 3′,3′-cGAMP. a**, Section of a representative electron micrograph (*n* = 1,148 from 1 dataset) of *Sf*STING oligomers formed in the presence of 3′,3′-cGAMP. **b**, Data processing scheme used to generate **c**. **c**, Reconstructions of *Sf*STING complexes bound to 3′,3′-cGAMP, coloured by local resolution, masked and unmasked. **d**, Model of full-length *Sf*STING bound to 3′,3′-cGAMP which appears to have additional TIR to TIR suprafilament contacts (linking one filament to another). Insufficient resolved density for the TIR domains in this processed dataset limits further analysis of this interaction. The final uploaded model does not contain the amino acids in the TIR domains for this reason. The STING to STING cross-filament 'tetramer' interface is similar to the c-di-GMP single filament model but filaments do not extend beyond an average of 4–6 units.

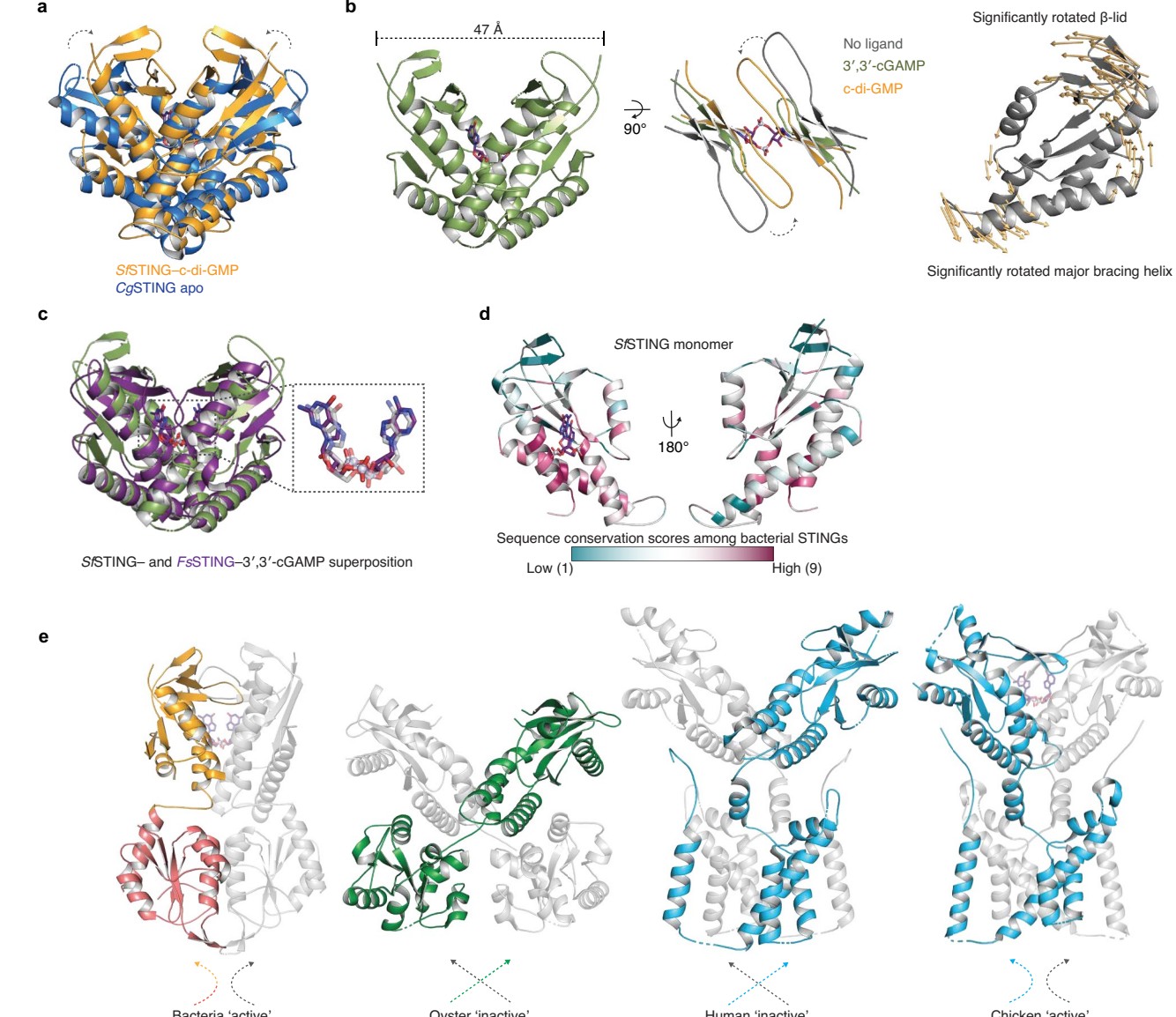

**Extended Data Fig. 4 | Comparison of cyclic dinucleotide-bound states of SfSTING. a**, c-di-GMP bound *Sf*STING superposition and comparison with apo *C. granulosa Cg*STING (PDB 6WT5)[4]. **b**, 3′,3′-cGAMP bound *Sf*STING CBD presents an intermediately compact state between the predicted apo (model prepared by superposing an *Sf*STING monomer with apo *Cg*STING PDB 6WT4 as guide) and the c-di-GMP bound structure (left, middle). Density is insufficient to accurately model the complete β-lid for the 3′,3′-cGAMP state. Panel at right presents a side view of a monomer of *Sf*STING and showcases the dynamic rotation/translation of each monomer relative to one another that sequesters the cyclic dinucleotide within the binding pocket. Orange arrows plot the domain rotation from apo (model, grey) to c-di-GMP bound state (using Modevectors python script). Movement of one monomer shown for clarity. **c**, Comparison of 3′,3′-cGAMP with *Sf*STING to 3′,3′-cGAMP with *Flavobacteriaceae sp. Fs*STING (PDB 6WT4)[4]. Superposition RMSD of 2 Å with a sequence identity in the STING CBD of only 25% highlights the incredible

degree of structural conservation in this family of bacterial proteins and exemplifies the shared mechanism for cyclic dinucleotide recognition. **d**, Conservation of bacterial STING sequence plotted on the *Sf*STING monomeric STING CBD domain. Conservation based on multiple sequence alignment of 102 bacterial STING CBD sequences. Analysis and colouring done using the ConSurf webserver[36]. Conservation scores range from 1 to 9 with increasing conservation. Highest conservation is evident within the core CDN binding pocket and along the dimerization interface. Weak conservation is found on the surface and at predicted filament contact sites. **e**, *Sf*STING structural comparison with the apo structure of a TIR-STING homolog from *Crassostrea gigas* (PDB 6WT6) and cryo-EM structures of human (PDB 6NT5) and chicken (PDB 6NT8) STING in the apo and 2′,3′-cGAMP-bound states highlights differences in linker crossover and domain packing. Portions of STING CBD structures omitted for clarity.

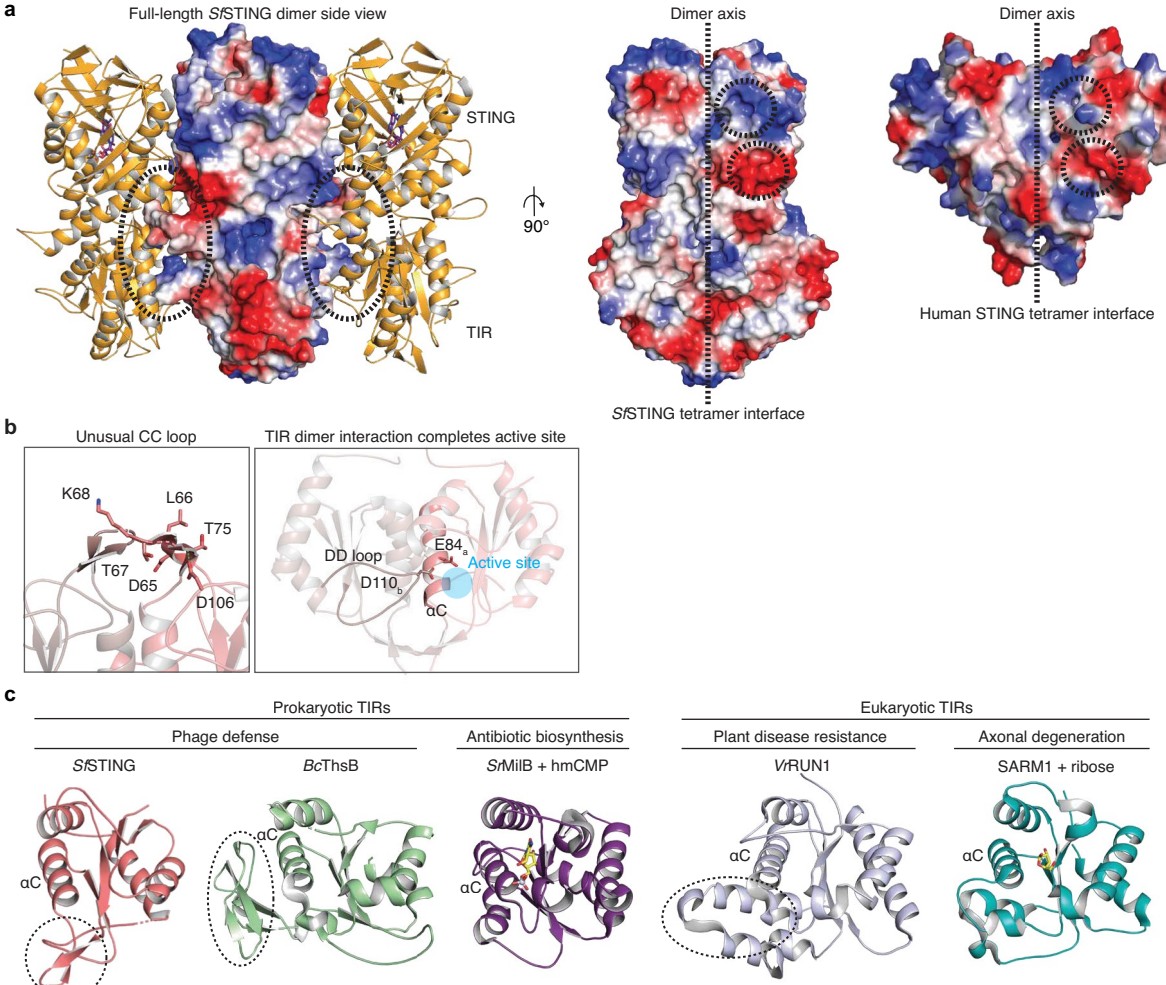

**Extended Data Fig. 5 | Structural analysis of the *Sf*STING TIR NADase domain. a**, Surface electrostatic potential for the core dimer of *Sf*STING with c-di-GMP single fibre filament (left). Colouring represents surface charges (red-negative, blue-positive, white-neutral). Mainly net-neutral charged loop projections mediate filament contacts (left). Dimer electrostatics at the tetramer filament interface for *Sf*STING and human STING (middle and right, respectively). Conserved patches of negative and positive charge are circled. **b**, Alternative views of the TIR dimerization interface. The TIR CC loop βD' and βE' insertion in the TIR domain of *Sf*STING (left) with symmetry related side chains omitted for clarity. Residue D110 from the DD loop is positioned by filament contacts to insert directly into the active site of the opposing symmetry mate (right). **c**, Global comparison of *Sf*STING TIR domain monomer with prokaryotic *Bc*ThsB (*Bacillus cereus* MSX-D12 TIR from the Thoeris phage defense system, PDB 6LHY)[20]; *Sr*MilB (*Streptomyces rimofaciens* hmCMP glycosylhydrolase PDB 4OHB)[37]; and eukaryotic TIRs *Vr*Run1 (*Vitis rotundifolia* plant NLR important for resistance to grapevine disease, PDB 7RX1)[21]; SARM1 (*Homo sapiens* TIR domain containing protein important for axonal degeneration, PDB 6O0Q)[19]. αC denoted for orientation and because it contains the catalytic glutamate residue in each structure. While all TIR-like proteins generally share a common fold, dashed regions indicate unique and poorly conserved structural elements found within several characterized active hydrolases.

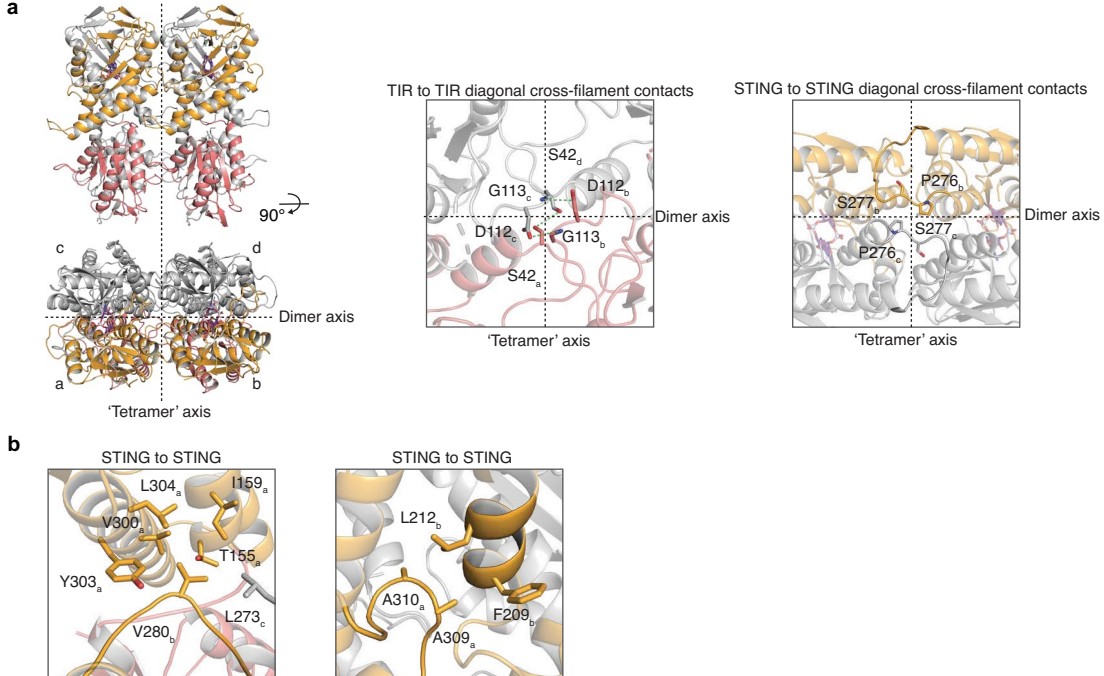

**Extended Data Fig. 6 | Structural analysis of the *Sf*STING cross-filament contacts. a**, Diagonal cross-filament contacts. *Sf*STING filaments are organized along mainly two planes, one separating pseudosymmetric dimer mates and one at the interface between 'tetrameric' dimers (left). Diagonal cross-filament interactions are those that cross the 'tetramer' and dimer planes. Additional views of TIR to TIR (middle, STINGs removed for clarity) and STING to STING (right, TIRs removed for clarity) contacts not detailed in main text. S42 is on the TIR BB loop and D112 and G113 are on the TIR DD loop. The STING loops containing P276 and S277 do not make direct electrostatic contacts but do make up a nearly continuous surface that effectively links the cross-tetramer plane related mates. **b**, Hydrophobic STING to STING cross-filament interactions. The non-polar side chain of V280 extends from a mostly polar and unstructured loop into a hydrophobic cavity formed across the filament 'tetramer' interface (left). A309 and A310 brace the hydrophobic face of an opposing α-helix from a STING 'tetramer' mate (right).

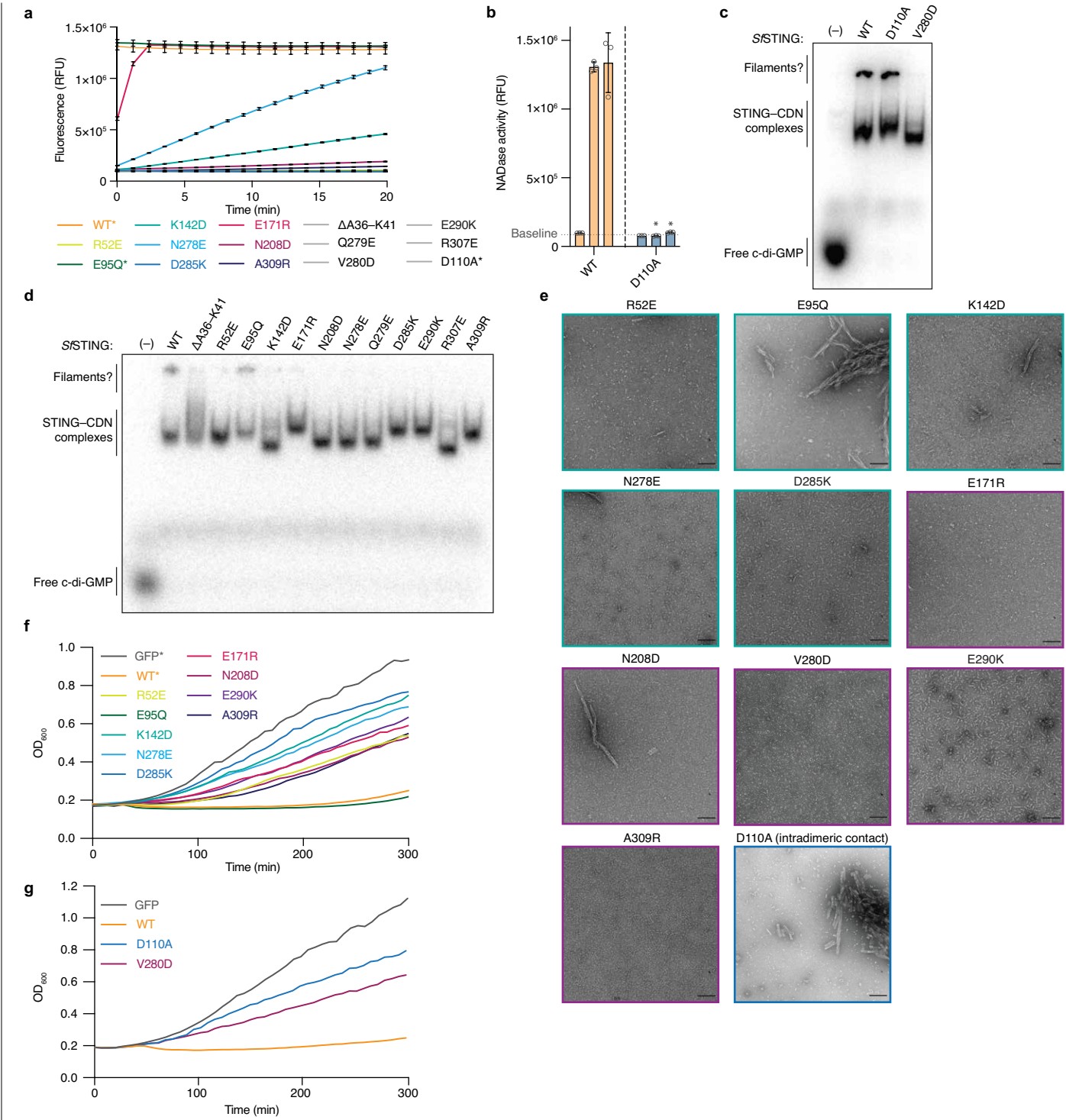

**Extended Data Fig. 7** | See next page for caption.

**Extended Data Fig. 7 | Mutational analysis of the *Sf*STING cross-filament contacts. a**, Raw plate reader NADase assay measuring fluorescence increase over time as a proxy for NAD⁺ degradation. At 1 µM enzyme, WT* and E95Q* mutant *Sf*STING consume the ε-NAD substrate within the deadtime of the experiment (flat maximal signal). Error bars indicate standard deviation for average of three technical replicates. ΔA36–K41, Q279E, V280D, E290K, R307E, and D110A* (grey) show no activity at this enzyme concentration, and lines are obscured by other inactive construct curves. D110A* indicates that this is not a filament contact mutant but an intradimeric contact mutant. **b**, Bar-graph representation of NADase activity for mutant D110A similar to presentation in Fig. 4a. NADase activity is measured as fluorescence intensity at 5 min. Each bar within a set corresponds to 0.1, 1, and 10 µM enzyme. Baseline threshold indicates background fluorescent signal. Error bars indicate standard deviation for average of three technical replicates. Data representative of three independent biological replicates. One-way ANOVA comparison of mutant to WT at the same protein concentration with statistically significant mean values ($P < 0.0001$) marked by * or otherwise are greater than $P = 0.05$ and considered not significantly different. **c**, EMSA showing radiolabelled c-di-GMP binding to WT, D110A, and V280D mutant *Sf*STING. Note: lack of probable filament band for the V280D mutant. Image representative of $n = 2$ experiments. **d**, EMSA showing radiolabelled c-di-GMP binding to all mutants. Note: lack of probable filament band for majority of mutants, consistent with lack of observed filaments by electron microscopy. **e**, Extended negative-stain EM micrographs for all mutants in the presence of c-di-GMP. Scale bar represents 100 nm. Images representative of $n \geq 3$ experiments. **f**, Extended plate reader growth curve assay data for most tested mutants. WT* and GFP* curves are identical to Fig. 4c. All curves represent the mean of 3 technical replicates. Data representative of >2 independent biological replicates. **g**, Plate reader growth curve assay for additional mutants V280D and D110A. All curves represent the mean of 3 technical replicates. Data representative of >2 independent biological replicates.

| | c-di-GMP single (EMD-26616) (PDB 7UN8) | c-di-GMP double (EMD-26617) (PDB 7UN9) | 3′,3′-cGAMP tetra (EMD-26619) | 3′,3′-cGAMP masked (EMD-26618) (PDB 7UNA) |
|---|---|---|---|---|
| **Data collection and processing** | | | | |
| Magnification | 105,000 | 105,000 | 36,000 | |
| Voltage (kV) | 300 | 300 | 200 | |
| Electron exposure (e–/Å$^2$) | 55.5 | 55.5 / 51.7 | 52.9 | |
| Defocus range (μm) | -1.2 to -2.1 | -1.2 to -2.1 / -1.2 to -2.5 | -1.4 to -2.6 | |
| Pixel size (Å) | 0.825 | 0.825 | 1.1 | |
| Symmetry imposed | C1 | C1 | C1 | |
| Initial particle images (no.) | 277,287 | 2,593,147 | 452,469 | |
| Final particle images (no.) | 26,447 | 355,128 | 105,567 | |
| Map resolution (Å) | 3.3 | 3.3 | 4.3 | 4.0 |
| FSC threshold | 0.143 | 0.143 | 0.143 | 0.143 |
| Map resolution range (Å) | 3.1 to 5.7 | 3.1 to 4.9 | 3.7 to 18.2 | 3.7 to 7.3 |
| | | | | |
| **Refinement** | | | | |
| Initial model used (PDB code) | 6WT5 | c-di-GMP single | | c-di-GMP single |
| Model resolution (Å) | 3.4 | 3.4 | | 4.2 |
| FSC threshold | 0.5 | 0.5 | | 0.5 |
| Model resolution range (Å) | 28-3.4 | 33-3.4 | | 49-4.2 |
| Map sharpening *B* factor (Å$^2$) | -34.3 | -90.6 | | -80 |
| Model composition | | | | |
| Non-hydrogen atoms | 14,067 | 17,675 | | 9,211 |
| Protein residues | 1,780 | 2,244 | | 1,132 |
| Ligands | 3 | 6 | | 4 |
| *B* factors (Å$^2$) | | | | |
| Protein | 59.7 | 56.5 | | 76.9 |
| Ligand | 46.3 | 44.1 | | 68.1 |
| R.m.s. deviations | | | | |
| Bond lengths (Å) | 0.004 | 0.003 | | 0.003 |
| Bond angles (°) | 0.811 | 0.522 | | 0.591 |
| Validation | | | | |
| MolProbity score | 2.06 | 1.61 | | 2.09 |
| Clashscore | 13.44 | 7.7 | | 13.42 |
| Poor rotamers (%) | 0 | 0.05 | | 0 |
| Ramachandran plot | | | | |
| Favored (%) | 93.5 | 96.9 | | 92.9 |
| Allowed (%) | 6.3 | 3.1 | | 7.1 |
| Disallowed (%) | 0.2 | 0 | | 0 |

**Extended Data Fig. 8 | Cryogenic-electron microscopy data summary table.** Table containing details of cryo-EM data collection, processing, and refinement including relevant statistics for all maps and models generated in this study.

# Reporting Summary

## Statistics

For all statistical analyses, confirm that the following items are present in the figure legend, table legend, main text, or Methods section.

| n/a | Confirmed | |
|---|---|---|
| ☐ | ☒ | The exact sample size (*n*) for each experimental group/condition, given as a discrete number and unit of measurement |
| ☒ | ☐ | A statement on whether measurements were taken from distinct samples or whether the same sample was measured repeatedly |
| ☐ | ☒ | The statistical test(s) used AND whether they are one- or two-sided *Only common tests should be described solely by name; describe more complex techniques in the Methods section.* |
| ☒ | ☐ | A description of all covariates tested |
| ☒ | ☐ | A description of any assumptions or corrections, such as tests of normality and adjustment for multiple comparisons |
| ☐ | ☒ | A full description of the statistical parameters including central tendency (e.g. means) or other basic estimates (e.g. regression coefficient) AND variation (e.g. standard deviation) or associated estimates of uncertainty (e.g. confidence intervals) |
| ☐ | ☒ | For null hypothesis testing, the test statistic (e.g. *F*, *t*, *r*) with confidence intervals, effect sizes, degrees of freedom and *P* value noted *Give P values as exact values whenever suitable.* |
| ☒ | ☐ | For Bayesian analysis, information on the choice of priors and Markov chain Monte Carlo settings |
| ☒ | ☐ | For hierarchical and complex designs, identification of the appropriate level for tests and full reporting of outcomes |
| ☒ | ☐ | Estimates of effect sizes (e.g. Cohen's *d*, Pearson's *r*), indicating how they were calculated |

*Our web collection on statistics for biologists contains articles on many of the points above.*

## Software and code

Policy information about availability of computer code

| | |
|---|---|
| Data collection | SerialEM 3.8.5 and 3.8.6 |
| Data analysis | Phenix 1.17, Coot 0.8.9, PyMOL 2.5.1, GraphPad Prism 9.3.0, RELION 3.1, cryoSPARC 3.1.018, Chimera 1.2.5, MotionCor2 1.4.0, GCTF 1.06 |

For manuscripts utilizing custom algorithms or software that are central to the research but not yet described in published literature, software must be made available to editors and reviewers. We strongly encourage code deposition in a community repository (e.g. GitHub). See the Nature Portfolio guidelines for submitting code & software for further information.

## Data

Policy information about availability of data

All manuscripts must include a data availability statement. This statement should provide the following information, where applicable:

- Accession codes, unique identifiers, or web links for publicly available datasets
- A description of any restrictions on data availability
- For clinical datasets or third party data, please ensure that the statement adheres to our policy

Data that support the findings of this study are available within the article and its Extended Data, Supplementary Data, and Source Data. Protein Data Bank (PDB) database accessions are listed where appropriate. All generated cryo-EM structure data have been deposited to the Protein Data Bank (PDB) database and will be made available upon public release of this manuscript. PDB accession codes are as follows: SfSTING single fibers with c-di-GMP- 7UN8 and EMD-26616, SfSTING double fibers with c-di-GMP- 7UN9 and EMD-26617, SfSTING short fibers with 3',3'-cGAMP (masked)- 7UNA and EMD-26618, and SfSTING short fibers with 3',3'-cGAMP- EMD-26619. PDB 6WT5 was used as a starting model for docking and building the SfSTING structure into the cryo-EM maps.

# Field-specific reporting

Please select the one below that is the best fit for your research. If you are not sure, read the appropriate sections before making your selection.

☒ Life sciences    ☐ Behavioural & social sciences    ☐ Ecological, evolutionary & environmental sciences

For a reference copy of the document with all sections, see nature.com/documents/nr-reporting-summary-flat.pdf

# Life sciences study design

All studies must disclose on these points even when the disclosure is negative.

| Sample size | No sample size calculation was performed and this generally is not relevant for biochemical and bacterial growth experiments. Bacterial cultures were grown to optical density (OD600) within linear range of absorbance measuring instrumentation. |
|---|---|
| Data exclusions | No data were excluded from the analyses. |
| Replication | All experiments were performed with independent replicates as described and based on widely adopted protocols in the field (2-3 technical replicates per experiment, >2 independent biological replicates). All attempts at replication were successful within expected variation. |
| Randomization | Randomization is not relevant to the biochemical, structural, and bacterial growth experiments described in this work as this would not impact the interpretation of the results collected. |
| Blinding | Data were not blinded. Blinding is not relevant to the experiments described in this work as subjective analyses were not performed and blinding would not impact the interpretation of the results collected. |

# Reporting for specific materials, systems and methods

We require information from authors about some types of materials, experimental systems and methods used in many studies. Here, indicate whether each material, system or method listed is relevant to your study. If you are not sure if a list item applies to your research, read the appropriate section before selecting a response.

## Materials & experimental systems

| n/a | Involved in the study |
|---|---|
| ☒ | ☐ Antibodies |
| ☒ | ☐ Eukaryotic cell lines |
| ☒ | ☐ Palaeontology and archaeology |
| ☒ | ☐ Animals and other organisms |
| ☒ | ☐ Human research participants |
| ☒ | ☐ Clinical data |
| ☒ | ☐ Dual use research of concern |

## Methods

| n/a | Involved in the study |
|---|---|
| ☒ | ☐ ChIP-seq |
| ☒ | ☐ Flow cytometry |
| ☒ | ☐ MRI-based neuroimaging |

