## [Peer Review File · Nature]

Manuscript Title: Cryo-EM structure of an active bacterial TIR-STING filament complex

Reviewer Comments & Author Rebuttals

Reviewer Reports on the Initial Version:

Referees' comments:

Referee #1 (Remarks to the Author):

Morehouse et al. determine multiple structures of bacterial TIR-STING filaments assembled with a strongly activating secondary messenger 3'3'-c-di-GMP, and a weakly activating 3'3'-c-GMP. These structures reveal conformational changes driven by secondary messenger binding that govern filament formation and ultimately NAD⁺ degradation. The work has been performed to a high standard and the model that they derive from the data presented (and other relevant literature) is solid. I recommend this manuscript for publication, provided the following issues are addressed.

Major points

1. A major finding of the study is the conformational change in STING induced by 3'3'-c-di-GMP binding that enables filament formation, and 3'3'-c-GMP induces a more moderate conformational change. Could any of these hypotheses related to structural changes be tested in the in vitro assays used in Fig 4? The manuscript would be greatly strengthened if the model (based on the structures) was supported by functional data.
2. It is unclear how the parallel vs. antiparallel dimer organization in Fig 2 fits into the model. How is this conformational change achieved? Based on the structural comparisons in Fig 2, it seems like a major conformational change, and quite distinct from what is shown in Fig 1. The schematic in Fig 4 does not appear to reflect this data. If the parallel vs. antiparallel observation in Fig 2 is merely interesting, perhaps it should be moved to the supplementary data.
3. Conservation of residues involved in protein-protein interfaces are suggested to be conserved based on structures of bacterial and human STING. To substantiate these claims, conservation of said residues within other STING homologues is necessary. If the conservation is as significant as the authors claim, then this should be apparent based on multiple sequence alignment with other related STING proteins. This additional analysis should be accompanied with a ConSurf depiction (Extended Data). It would be great to show a STING monomer colored by conservation and highlight conserved monomer-monomer interfaces (as well as other functionally important regions of STING).

Minor points

1. For conformational changes, I strongly encourage the authors to show modevectors to depict the change in each C-alpha position driven by 3'3'-c-di-GMP binding. This applies to Fig 1b and Extended Data Fig 4. This would make the story much easier to follow than overlaying two busy cartoon depictions.

2. In Fig 1b & c the 'unbound' structure is grey, and the 'bound' structure is gold...but the inside is grey? In panel c, it is unclear whether the transparent protein cartoon structure is an overlay of both models or simply the active one. The inside and outside of the helices of each cartoon model should be consistent for the sake of clarity.

David Taylor

Referee #2 (Remarks to the Author):

STING proteins are cyclic dinucleotide-activated sensors that mediate antiviral signaling in both metazoan innate immunity and bacterial phage defense pathways. This study reports the cryo-EM structure of a bacterial TIR-STING filament complex, revealing that cyclic-di-GMP drives oligomeric filament formation through conserved interactions similar to those observed for human STING. Filament formation results in the activation of the NADase activity of the TIR domains to trigger abortive infection, as validated by *in vivo* assays. These results provide the structural basis for filament formation in STING signaling, highlighting the conservation of the pathway between prokaryotes and metazoan innate immunity.

Overall, this is a timely and important study that provides novel insights into the mechanism of prokaryotic STING homologs in anti-phage immunity and raises unexpected mechanistic parallels between prokaryotic and metazoan STING pathways. The study is likely to be of high interest for the innate immunity and prokaryotic genome defense fields.

I would have the following question/comments for the revision of the manuscript:

1. What is meant by "wrapped double filament" (p. 3, line 27)? The meaning is not entirely clear to me. It would be helpful if the authors could provide a better description of the single and double filaments, perhaps in terms of the symmetry operations (translation & rotation) that relate adjacent protomers within the filaments.
2. P. 3, lines 47-48. Refer to fig. 1a here?
3. P. 4, line 59-60. Please correct sentence. "The active state conformation of *G. gallus* STING..."
4. P. 5, line 81. What is meant by "head-to-tail" oligomerization? In fig. 1a & 3a, the protomers appear to be parallel. Would this not be lateral "head-to-head"?
5. P. 3, line 35. "Compared to the modeled open apo state...". How was the open state modeled?
6. How is the oligomerization affected by 3'3'-cGAMP binding so as to terminate filament formation after 4-6 units?
7. What is the function of D110 in the loop that completes the active site upon TIR domain dimerization? Wouldn't TIR dimerization be expected to be sufficient for activity? How does filament formation affect the kinetics of the NADase activity?

Referee #3 (Remarks to the Author):

Nature paper review, 2/16/2022

Stimulator of Interferon Genes (STING) is an evolutionally conserved family of cyclic dinucleotides (CDNs) sensors that mediate innate antiviral responses in metazoan and antiphage responses in bacteria. Mammalian STING is a membrane protein localized on the ER membrane. Upon ligand binding, they oligomerize and serve as a structural framework for the recruitment and activation of downstream protein kinase TBK1 and transcription factor IRF-3, which regulates the induction of type I interferons. Unlike metazoan STING, bacterial STING contains a TIR domain at its N-terminus. Upon ligand binding, bacterial STING oligomerize and mediate the activation of the NADase activity of the TIR domains, which hydrolyse NAD⁺ and restrict the replication of bacterial phages. The molecular mechanism of metazoan STING activation upon ligand binding had been extensively studied. However, the structural basis of bacterial STING activation remains poorly understood. In this study, Morehouse et al. presented the first high resolution cryo-EM structure of bacterial STING (referred to as SfSTING hereafter) oligomers in complex with c-di-GMP, which reveals a structurally conserved mode of STING activation through side-by-side stacking of bacterial STING dimers. The authors also conducted extensive mutagenesis and functional studies to demonstrate that ligand binding induced oligomerization of SfSTING is critical for the activation of the NADase activity of the TIR domain. Overall, this is a very interesting and exciting study, which reveals the molecular basis of bacterial STING activation by CDN. My major concern about this manuscript is that it was obviously a rushed draft likely due to unfavorable competitions. The manuscript needs to be carefully polished to be considered for publication.

Here are some issues about the manuscript:

1. It may be more appropriate to use SfSTING instead of TIR-STING in many occasions. The mixed usage of these two different names is a little confusing.

2. The oligomerization of vertebrate STING upon cGAMP binding is a major step in STING mediated signaling. The oligomerization of STING was first observed by fluorescence microscopy. Recent studies by Ergun et al. (Cell, 2019) showed that the cyclic dinucleotide binding domain of STING form side-by-side oligomers in the crystal lattice. A recent study by Shang et al. (Nature, 2019) observed that full-length chicken STING dimers forms a side-by-side tetramer at 6.5 Angstrom resolution. Although a model of STING oligomerization upon cGAMP binding was proposed, no experimental evidence was provided to demonstrate the formation of higher order STING oligomers as presented in this work. Another study by Zhang et al. (Nature, 2019) provided a structural model of STING oligomer that is likely involved in TBK1 recruitment. Again, no experimental evidence about ligand binding induced oligomerization of STING was presented. The direct evidence of cGAMP induced oligomerization of full-length human STING was presented by cryo-EM studies by Zhao et al. (Nature, 2019). The formation of bacterial STING oligomers was also shown previously by Morehouse et al. (Nature, 2020). No high-resolution structure of ligand bound STING oligomers is available to date. Considering only 18 papers were cited in this manuscript, I wish the authors did not overlook some of the important studies in the field and provide an accurate and balanced overview of the literature, which should greatly increase the significance of the current study.

3. Line 16. The mechanism of mammalian STING activation and signaling is quite different from bacterial STING (see also the first paragraph of this comments). It is more appropriate and accurate to state that this study defines the structural basis of prokaryotic STING activation.

4. It will be nice to use the same terminologies for CDNs, such as c-di-AMP, c-di-GMP, 2',3' cGAMP, and 3', 3' cGAMP, as in the previous studies (Morehouse et al. Nature, 2019) and in other papers.

5. Is the SfSTING/c-di-GMP complex sample soluble? what is the average mass? How many STING dimers are included in the complex? These information can be provided before describing the EM structures to show how the samples were prepared.

6. The authors mentioned in multiple occasions the high affinity binding of c-di-GMP to SfSTING. However, the binding affinities of CDNs to SfSTING or its ligand binding domain were not presented in the manuscript. It is not clear which ligand binds at higher affinity and which one is functionally relevant.

7. Two kinds of SfSTING fibrils were observed in this study. The single fibril structure presented only accounts for about 26K particles, while the double fibril structure accounts for 355K particles. It was not clear why the double fibril structure was not discussed in depth. Does that structure contain additional intermolecular contacts that may be functionally relevant?

8. It seems more appropriate to describe the oligomerization of SfSTING dimers as side-by-side rather than head-to-tail stacking of the STING dimers.

9. Line 30. What is the buried surface area described here? Is it between two STING dimers, or between a STING dimer and two of its neighboring dimers?

10. Line 31. To be active, the STING oligomer may not be rigid. It should be able to bind NAD⁺ and catalysis the hydrolysis of the substrate.

11. Line 55-62. It seems that ligand binding induces a major conformation change of SfSTING. It will be great to show the density for individual STING dimers in the filament and density of the linker region between the TIR domain and the ligand binding domains.

12. Why the cGAMP bound structure not discussed in more detail? Is it functionally relevant? From their previous studies, it seems that only c-di-GMP activate SfSTING (Morehouse et al, Nature 2019). If cGAMP is not a functionally relevant ligand of SfSTING, the structure may not be included in this manuscript.

13. It is likely that ligand binding induced oligomerization and conformation change of the TIR domains mediates the activation of SfSTING. However, it was not clearly shown what kind of conformation change happened in the TIR domain that activates the NADase. Comparison of TIR domain structures in the active and inactive states, for example the current structure and the oyster STING structure, may provide some clue. Superposition of SfSTING TIR domain structure with the recent RUN1 TIR bound to NAD structure (Burdett et al, Biorxiv, 2021, PDB, 7S2Z) may reveal where

the active sites are located and if they are accessible by NAD⁺ in SfSTING fibril.

14. The mutagenesis and functional assay data in Figure 4a are not so informative without statistical analysis. It will be better to show the data as in ED Figure 7b. Statistical analysis should be included to show the quality of the data and how the mutants compare with the WT.

15. I am not sure if it was appropriate to cite the Hogrel manuscript without providing a little more details.

Author Rebuttals to Initial Comments:

Referee #1:

Morehouse et al. determine multiple structures of bacterial TIR-STING filaments assembled with a strongly activating secondary messenger 3'3'-c-di-GMP, and a weakly activating 3'3'-c-GMP. These structures reveal conformational changes driven by secondary messenger binding that govern filament formation and ultimately NAD⁺ degradation. The work has been performed to a high standard and the model that they derive from the data presented (and other relevant literature) is solid. I recommend this manuscript for publication, provided the following issues are addressed.

We thank the reviewer for recommending our manuscript for publication and we appreciate their helpful feedback to improve presentation of our data.

Major points

1. A major finding of the study is the conformational change in STING induced by 3'3'-c-di-GMP binding that enables filament formation, and 3'3'-c-GMP induces a more moderate conformational change. Could any of these hypotheses related to structural changes be tested in the in vitro assays used in Fig 4? The manuscript would be greatly strengthened if the model (based on the structures) was supported by functional data.

Our previous biochemical results demonstrate c-di-GMP-induced SfSTING signaling is >1,000× more potent than 3',3'-cGAMP (Morehouse et al Nature 2020 PMID 32877915), and we therefore have focused our experimental analysis on validating contacts observed in the SfSTING–c-di-GMP filament structure. c-di-GMP recognition results in closure of the SfSTING lid domain and a conformational change that exposes surface contact sites that create an interface for nucleating STING filament formation (Fig. 3). To validate these structural observations, we tested a series of 13 mutations predicted to disrupt cross-filament contacts. Importantly, our mutational analysis demonstrates that disruption of cross-filament contacts abolishes signaling and uncouples c-di-GMP recognition from TIR-domain NADase activity. These results provide functional support for our model of STING filament formation (Fig. 4d). As suggested by Reviewer #3, we have clarified these important findings by moving the primary structural mutagenesis data into main text Fig. 4a.

To further address the reviewer's point, we now provide additional experimental data testing two new SfSTING mutants D110A and V280D (see new Extended Data Fig. 7). Residue V280 in the SfSTING CDN binding domain forms part of a hydrophobic interface buried upon SfSTING filament formation. In agreement with our previous analysis of cross-filament contacts, we observed that mutation of this residue abolishes SfSTING filament formation and NADase activity but does not impair c-di-GMP recognition. SfSTING residue D110 is part of a loop that reaches across the protomer interface to complete the NADase active site (Fig. 3d, Extended Data Fig. 5b). In contrast to cross-filament contacts, our SfSTING–c-di-GMP filament structure predicts that residue D110 is important for TIR domain NADase activity but not STING filament formation. Consistent with this prediction, we observe that SfSTING D110A protein retains the ability to recognize c-di-GMP and form long protein filaments but loses all ability to initiate final NADase activity (Extended Data Fig. 7). Together, our mutagenesis results validate individual steps of SfSTING signaling and support a complete model of bacterial STING activation.

2. It is unclear how the parallel vs. antiparallel dimer organization in Fig 2 fits into the model. How is this conformational change achieved? Based on the structural comparisons in Fig 2, it seems like a major conformational change, and quite distinct from what is shown in Fig 1. The schematic in Fig 4 does not appear to reflect this data. If the parallel vs. antiparallel observation in Fig 2 is merely interesting, perhaps it should be moved to the supplementary data.

We apologize for the confusion within our previous Figure 2. No specific evidence exists to suggest bacterial STING proteins undergo a parallel to antiparallel transition. However, as this structural transition has been a focus of human STING research (Shang et al Nature 2019 PMID 30842659; Lu et al Nature 2022 PMID 35388221) we included this panel to enable direct analysis of the active-state bacterial STING structure. We agree with the reviewer that this point is better focused for experts in the field and have moved the analysis to Extended Data Fig. 4 as suggested.

3. Conservation of residues involved in protein-protein interfaces are suggested to be conserved based on structures of bacterial and human STING. To substantiate these claims, conservation of said residues within other STING homologues is necessary. If the conservation is as significant as the authors claim, then this should be apparent based on multiple sequence alignment with other related STING proteins. This additional analysis should be accompanied with a ConSurf depiction (Extended Data). It would be great to show a STING monomer colored by conservation and highlight conserved monomer-monomer interfaces (as well as other functionally important regions of STING).

Bacterial and metazoan STING sequences share <10% sequence identity but exhibit a remarkable degree of structural homology. We previously used a structure-guided multiple sequence alignment to identify conserved features shared between the cyclic dinucleotide binding pockets of bacterial and metazoan STING proteins (Morehouse et al Nature 2020 PMID 32877915). The level of homology is much higher when comparing bacterial STING sequences alone, mainly in the CDN binding pocket and dimerization interface. To clarify conservation within the filament interfaces among bacterial STING

sequences we now include monomeric SfSTING cyclic dinucleotide binding domain structure colored according to prokaryotic STING sequence conservation (Extended Data Fig. 4d).

Minor points

1. For conformational changes, I strongly encourage the authors to show modevectors to depict the change in each C-alpha position driven by 3'3'-c-di-GMP binding. This applies to Fig 1b and Extended Data Fig 4. This would make the story much easier to follow than overlaying two busy cartoon depictions.

We thank the reviewer for this suggestion. We have modeled the conformational change using modevectors and now include this analysis within Extended Data Fig. 4. Our emphasis in Figure 1b is on the inward movement of a small portion of the structure and we therefore prefer to leave the image with shading highlights as originally presented.

2. In Fig 1b & c the 'unbound' structure is grey, and the 'bound' structure is gold...but the inside is grey? In panel c, it is unclear whether the transparent protein cartoon structure is an overlay of both models or simply the active one. The inside and outside of the helices of each cartoon model should be consistent for the sake of clarity.

We have elected to retain the coloring scheme to be consistent with our previous crystal structural analysis of bacterial STING cyclic dinucleotide interactions (Morehouse et al Nature 2020 PMID 32877915). However, we appreciate the reviewer for pointing out this potential source of confusion and we have clarified the figure legend to define individual STING protomers more clearly in Fig. 1c.

David Taylor

We thank the reviewer again for their helpful comments to improve our manuscript.

Referee #2:

STING proteins are cyclic dinucleotide-activated sensors that mediate antiviral signaling in both metazoan innate immunity and bacterial phage defense pathways. This study reports the cryo-EM structure of a bacterial TIR-STING filament complex, revealing that cyclic-di-GMP drives oligomeric filament formation through conserved interactions similar to those observed for human STING. Filament formation results in the activation of the NADase activity of the TIR domains to trigger abortive infection, as validated by in vivo assays. These results provide the structural basis for filament formation in STING signaling, highlighting the conservation of the pathway between prokaryotes and metazoan innate immunity.

Overall, this is a timely and important study that provides novel insights into the mechanism of prokaryotic STING homologs in anti-phage immunity and raises unexpected mechanistic parallels between prokaryotic and metazoan STING pathways. The study is likely to be of high interest for the innate immunity and prokaryotic genome defense fields.

We thank the reviewer for highlighting our work as of high interest to multiple fields and for their helpful comments to improve our manuscript.

I would have the following question/comments for the revision of the manuscript:

1. What is meant by "wrapped double filament" (p. 3, line 27)? The meaning is not entirely clear to me. It would be helpful if the authors could provide a better description of the single and double filaments, perhaps in terms of the symmetry operations (translation & rotation) that relate adjacent protomers within the filaments.

We thank the reviewer for pointing out this issue. We have deleted this term and modified the text descriptions to clearly define the double filament structure as follows "...and antiparallel double filament structures that make supra-molecular contacts between STING and TIR domains of opposing filaments (Extended Data Fig. 1, 2). The TIR domains are not as well resolved in the major double filament class likely due to conformational heterogeneity and we therefore focused structural analysis on the single fiber filaments. A 3.3 Å cryo-EM reconstruction of the dominant class of single-fiber filaments reveals SfSTING oligomerizes through formation of a repeating laterally translated array of parallel stacked protein dimers that buries >3,000 Å² of surface area between two pairs of dimers and locks the STING cyclic dinucleotide (CDN) binding domain and associated TIR effector domains into filamentous assemblies capable of reaching >300 nm in length (Fig. 1a)."

2. P. 3, lines 47-48. Refer to fig. 1a here?

We have modified the text as suggested.

3. P. 4, line 59-60. Please correct sentence. "The active state conformation of G. gallus STING..."

We have corrected the sentence as follows: “The active state conformation of the *G. gallus* STING...”

4. P. 5, line 81. What is meant by “head-to-tail” oligomerization? In fig. 1a & 3a, the protomers appear to be parallel. Would this not be lateral “head-to-head”?

We have modified the text to specifically refer to the filament organization as lateral head-to-head.

5. P. 3, line 35. “Compared to the modeled open apo state...”. How was the open state modeled?

The open state was modeled using superposition with a previously determined crystal structure of an apo bacterial STING domain and this is now described in the figure legend for both Fig. 1 and Extended Data Fig. 4.

6. How is the oligomerization affected by 3',3'-cGAMP binding so as to terminate filament formation after 4-6 units?

Compared to the fully active SfSTING–c-di-GMP filament structure, 3',3'-cGAMP recognition induces impartial closure of the lid domain and an overall conformation in the STING CDN binding domain that likely leads to weakened or fewer interface contact sites than observed for c-di-GMP induced filaments. Additionally, a lack of well-defined density for the TIR domains in the 3',3'-cGAMP filaments suggests conformational flexibility that may impact stability of the filaments. We now describe these features in the main text.

7. What is the function of D110 in the loop that completes the active site upon TIR domain dimerization? Wouldn't TIR dimerization be expected to be sufficient for activity? How does filament formation affect the kinetics of the NADase activity?

Our mutagenesis analysis of cross-filament contacts in the active SfSTING–c-di-GMP structure demonstrates that filament formation is essential for TIR domain NADase activity. Single point mutations that disrupt these interfaces abolish all signaling and uncouple c-di-GMP recognition from TIR-domain NADase activity (Fig. 4 and Extended Data Fig. 7).

To further analyze TIR NADase domain activation, as new data in our revised manuscript we now specifically test the function of SfSTING D110A mutant protein (Extended Data Fig. 7). SfSTING residue D110 is part of a loop that reaches across the protomer interface to complete the NADase active site (Fig. 3d, Extended Data Fig. 5b). In contrast to cross-filament contacts, our SfSTING–c-di-GMP filament structure predicts that residue D110 is important for TIR domain NADase activity but not STING filament formation. Consistent with this prediction, we observe that SfSTING D110A protein retains the ability to recognize c-di-GMP and form long protein filaments but loses all ability to initiate final NADase activity (Extended Data Fig. 7). Together, our mutagenesis results validate individual steps of SfSTING signaling and support a complete model of bacterial STING activation. The structure of full-length SfSTING in the apo inactive conformation remains unknown. We agree that further analysis of this structural state and mechanisms preventing aberrant activation of TIR NADase-activity through TIR domain dimerization remain an interesting goal for future research.

Referee #3:

Nature paper review, 2/16/2022

Stimulator of Interferon Genes (STING) is an evolutionally conserved family of cyclic dinucleotides (CDNs) sensors that mediate innate antiviral responses in metazoan and antiphage responses in bacteria. Mammalian STING is a membrane protein localized on the ER membrane. Upon ligand binding, they oligomerize and serve as a structural framework for the recruitment and activation of downstream protein kinase TBK1 and transcription factor IRF-3, which regulates the induction of type I interferons. Unlike metazoan STING, bacterial STING contains a TIR domain at its N-terminus. Upon ligand binding, bacterial STING oligomerize and mediate the activation of the NADase activity of the TIR domains, which hydrolyse NAD⁺ and restrict the replication of bacterial phages. The molecular mechanism of metazoan STING activation upon ligand binding had been extensively studied. However, the structural basis of bacterial STING activation remains poorly understood.

In this study, Morehouse et al. presented the first high resolution cryo-EM structure of bacterial STING (referred to as SfSTING hereafter) oligomers in complex with c-di-GMP, which reveals a structurally conserved mode of STING activation through side-by-side stacking of bacterial STING dimers. The authors also conducted extensive mutagenesis and functional studies to demonstrate that ligand binding induced oligomerization of SfSTING is critical for the activation of the NADase activity of the TIR domain. Overall, this is a very interesting and exciting study, which reveals the molecular basis of bacterial STING activation by CDN. My major concern about this manuscript is that it was obviously a rushed draft likely due to unfavorable competitions. The manuscript needs to be carefully polished to be considered for publication.

We thank the reviewer for highlighting our study as interesting and exciting. We appreciate their feedback and are grateful for their detailed comments to improve the clarity and framing of our manuscript.

Here are some issues about the manuscript:

1. It may be more appropriate to use SfSTING instead of TIR-STING in many occasions. The mixed usage of these two different names is a little confusing.

We thank the reviewer for this suggestion. We have corrected TIR-STING to SfSTING where possible, and only left a few instances where we refer to the domain architecture of the TIR-STING protein and not a specific species or where we refer to TIR-STING homologs conserved in metazoan animals.

2. The oligomerization of vertebrate STING upon cGAMP binding is a major step in STING mediated signaling. The oligomerization of STING was first observed by fluorescence microscopy. Recent studies by Ergun et al. (Cell, 2019) showed that the cyclic dinucleotide binding domain of STING form side-by-side oligomers in the crystal lattice. A recent study by Shang et al. (Nature, 2019) observed that full-length chicken STING dimers forms a side-by-side tetramer at 6.5 Angstrom resolution. Although a model of STING oligomerization upon cGAMP binding was proposed, no experimental evidence was provided to demonstrate the formation of higher order STING oligomers as presented in this work. Another study by Zhang et al. (Nature, 2019) provided a structural model of STING oligomer that is likely involved in TBK1 recruitment. Again, no experimental evidence about ligand binding induced oligomerization of STING was presented. The direct evidence of cGAMP induced oligomerization of full-length human STING was presented by cryo-EM studies by Zhao et al. (Nature, 2019). The formation of bacterial STING oligomers was also shown previously by Morehouse et al. (Nature, 2020). No high-resolution structure of ligand bound STING oligomers is available to date. Considering only 18 papers were cited in this manuscript, I wish the authors did not overlook some of the important studies in the field and provide an accurate and balanced overview of the literature, which should greatly increase the significance of the current study.

We apologize for any lack of appropriate citation to the literature, we agree with the reviewer that balanced citation of previous findings in the field is important to fully explain the significance of our study. The seminal findings of Ergun et al., Shang et al., and Zhang et al., were cited in the original version of our manuscript. To further increase the clarity of these citations we have inserted additional lines of text to highlight the importance of these studies (see Lines 20–25). Additionally, we now include specific citation of the Zhao et al study (Zhao et al. Nature 2019 PMID 31118511) as well as references to observation of STING puncta formation in cells (Ishikawa et al Nature 2009 PMID 19776740), electrophoresis analysis of STING multimeric complexes (Tanaka and Chen Sci Signaling 2012 PMID 22394562; Haag et al Nature 2018 PMID 29973723), and artificial activation of STING upon fusion to multimerization domains (Sun et al PNAS 2009 PMID 19433799).

3. Line 16. The mechanism of mammalian STING activation and signaling is quite different from bacterial STING (see also the first paragraph of this comments). It is more appropriate and accurate to state that this study defines the structural basis of prokaryotic STING activation.

We agree and have modified the text as suggested.

4. It will be nice to use the same terminologies for CDNs, such as c-di-AMP, c-di-GMP, 2',3' cGAMP, and 3', 3' cGAMP, as in the previous studies (Morehouse et al. Nature, 2019) and in other papers.

We thank the reviewer for this comment and have corrected the terminology for consistency with previous papers.

5. Is the SfSTING/c-di-GMP complex sample soluble? what is the average mass? How many STING dimers are included in the complex? These information can be provided before describing the EM structures to show how the samples were prepared.

We have expanded the methods section to further explain preparation of the SfSTING samples for cryo-EM analysis. Upon exposure to the activating ligand c-di-GMP, solutions of purified SfSTING immediately begin filament formation and become visibly cloudy. SfSTING filaments appear as void peaks when analyzed by SEC-MALS limiting the ability to accurately determine mass (Morehouse et al Nature 2020 PMID 32877915). In our EM analysis, SfSTING activation is observed as individual filaments that range in size with some filaments reaching >300 nm in length (~85 dimer copies, ~6.3 MDa). Particles selected for processing and high-resolution structural analysis include density for at least 5 SfSTING dimer copies.

6. The authors mentioned in multiple occasions the high affinity binding of c-di-GMP to SfSTING. However, the binding affinities of CDNs to SfSTING or its ligand binding domain were not presented in the manuscript. It is not clear which ligand binds at higher affinity and which one is functionally relevant.

We have modified the text to state that SfSTING binds to c-di-GMP with ~300 nM apparent affinity (K_d) and that low nM (~10s of nM) c-di-GMP is sufficient to initiate robust NADase activity (Morehouse et al Nature 2020 PMID 32877915). Our previous results demonstrate that 3',3'-cGAMP binds with slightly weaker affinity (~700 nM K_d) and is unable to induce

robust TIR NADase activity, findings that are now explained by our cryo-EM analysis demonstrating that c-di-GMP recognition is required for complete closure of the STING lid domain and filament nucleation.

7. Two kinds of SfSTING fibrils were observed in this study. The single fibril structure presented only accounts for about 26K particles, while the double fibril structure accounts for 355K particles. It was not clear why the double fibril structure was not discussed in depth. Does that structure contain additional intermolecular contacts that may be functionally relevant?

We have added new text to increase description of this complex as follows: "...and antiparallel double filament structures that make supra-molecular contacts between STING and TIR domains of opposing filaments (Extended Data Fig. 1, 2). The TIR domains are not as well resolved in the major double filament class likely due to conformational heterogeneity and we therefore focused structural analysis on the single fiber filaments."

8. It seems more appropriate to describe the oligomerization of SfSTING dimers as side-by-side rather than head-to-tail stacking of the STING dimers.

We agree and have modified the text to "lateral head-to-head" rather than head-to-tail.

9. Line 30. What is the buried surface area described here? Is it between two STING dimers, or between a STING dimer and two of its neighboring dimers?

We have modified the text to clarify that the buried surface area is at the interface between two STING dimers.

10. Line 31. To be active, the STING oligomer may not be rigid. It should be able to bind NAD⁺ and catalysis the hydrolysis of the substrate.

We agree and for clarity have removed the term rigid.

11. Line 55-62. It seems that ligand binding induces a major conformation change of SfSTING. It will be great to show the density for individual STING dimers in the filament and density of the linker region between the TIR domain and the ligand binding domains.

We thank the reviewer for this suggestion. The cryo-EM density for an individual STING dimer is now shown in Extended Data Fig. 2d, and the density of the linker region between TIR and STING ligand binding domain (res 144–176) is presented in Extended Data Fig. 2c.

12. Why the cGAMP bound structure not discussed in more detail? Is it functionally relevant? From their previous studies, it seems that only c-di-GMP activate SfSTING (Morehouse et al, Nature 2019). If cGAMP is not a functionally relevant ligand of SfSTING, the structure may not be included in this manuscript.

We agree that the SfSTING–3',3'-cGAMP structural data are less important and have therefore presented figures of this structure mostly in the Extended Data. However, we prefer to retain this structure in the manuscript as it enhances comparative analysis of how correct c-di-GMP ligand recognition directs filament formation and reveals that the TIR NADase domains are only ordered for catalysis in the fully active SfSTING–c-di-GMP filament conformation.

13. It is likely that ligand binding induced oligomerization and conformation change of the TIR domains mediates the activation of SfSTING. However, it was not clearly shown what kind of conformation change happened in the TIR domain that activates the NADase. Comparison of TIR domain structures in the active and inactive states, for example the current structure and the oyster STING structure, may provide some clue. Superposition of SfSTING TIR domain structure with the recent RUN1 TIR bound to NAD structure (Burdett et al, Biorxiv, 2021, PDB, 7S2Z) may reveal where the active sites are located and if they are accessible by NAD⁺ in SfSTING fibril.

We thank the reviewer for this suggestion. We include TIR domain structural comparisons in Fig. 2d and Extended Data Fig. 5c highlighting TIR domains from humans (SARM1), plants (RUN1), and bacteria (ThsB). We read the referenced Burdett et al pre-print carefully and by our understanding the suggested RUN1 structure (PDB 7S2Z) does not have NAD⁺ present in the active site despite co-crystallization which limits its utility for analysis. However, we agree that the analysis in this paper regarding TIR oligomerization is relevant to our study and have now included specific citation of their findings.

To further analyze TIR NADase domain activation, as new data in our revised manuscript we now specifically test the function of SfSTING D110A mutant protein (Extended Data Fig. 7). SfSTING residue D110 is part of a loop that reaches across the protomer interface to complete the NADase active site (Fig. 3d, Extended Data Fig. 5b). In contrast to cross-filament contacts, our SfSTING–c-di-GMP filament structure predicts that residue D110 is important for TIR domain NADase activity but not STING filament formation. Consistent with this prediction, we observe that SfSTING D110A protein retains

the ability to recognize c-di-GMP and form long protein filaments but loses all ability to initiate final NADase activity (Extended Data Fig. 7). Together, our mutagenesis results validate individual steps of *Sf*STING signaling and support a complete model of bacterial STING activation. The structure of full-length *Sf*STING in the apo inactive conformation remains unknown. We agree that further analysis of this structural state and mechanisms preventing aberrant activation of TIR NADase-activity through TIR domain dimerization remain an interesting goal for future research.

14. The mutagenesis and functional assay data in Figure 4a are not so informative without statistical analysis. It will be better to show the data as in ED Figure 7b. Statistical analysis should be included to show the quality of the data and how the mutants compare with the WT.

We agree with the reviewer and have revised Fig. 4 to include our mutagenesis data within the main text along with statistical analysis of the results.

15. I am not sure if it was appropriate to cite the Hogrel manuscript without providing a little more details.

We appreciate the reviewer's point but think that citation of the Hogrel et al manuscript is appropriate as their data support effector oligomerization as a shared feature of diverse CBASS anti-phage defense systems.

Reviewer Reports on the First Revision:

Referees' comments:

Referee #1 (Remarks to the Author):

The authors have thoughtfully and thoroughly addressed all of my concerns and incorporated most of the suggestions, which I think strengthens the manuscript. I recommend publication in Nature.

David Taylor

Referee #2 (Remarks to the Author):

My previous comments and concerns have been adequately addressed - I do not have any further criticisms.

I congratulate the authors on a great work.

Referee #3 (Remarks to the Author):

The authors have fully addressed questions raised in the previous review. The manuscript is clearly written and the results are nicely presented in the revised manuscript. I strongly support a timely publication of the revised manuscript.

I have a few minor suggestions that I hope the authors may consider during the final revisions of the manuscript:

1. Line 58, 59, please use KD instead of Kd for binding affinities.
2. L103, please check the geometry and distances between these residues. It may be more appropriate to describe these interactions as electrostatic interactions.
3. L238, please describe c-di-GMP as purple stick model.
4. L244, 264, I suggest close up view instead of magnified view in these figure legends.

Author Rebuttals to First Revision:

Referee #1:

The authors have thoughtfully and thoroughly addressed all of my concerns and incorporated most of the suggestions, which I think strengthens the manuscript. I recommend publication in Nature.

David Taylor

We thank the reviewer for recommending our manuscript for publication and we appreciate their helpful feedback to improve presentation of our data.

Referee #2 (Remarks to the Author):

My previous comments and concerns have been adequately addressed - I do not have any further criticisms.

I congratulate the authors on a great work.

We thank the reviewer for their kind words and consideration of our work.

Referee #3 (Remarks to the Author):

The authors have fully addressed questions raised in the previous review. The manuscript is clearly written and the results are nicely presented in the revised manuscript. I strongly support a timely publication of the revised manuscript.

We thank the reviewer for saying our manuscript is clearly written and nicely presented and thank them for their time and consideration of our revised work.

I have a few minor suggestions that I hope the authors may consider during the final revisions of the manuscript:

1. Line 58, 59, please use KD instead of Kd for binding affinities.

Following convention and for consistency with our previous paper on this topic (Morehouse et al Nature 2020), we prefer to use K_d (uppercase K, subscript and lowercase d) to define the dissociation constant. We have adjusted the text appropriately to reflect this change.

2. L103, please check the geometry and distances between these residues. It may be more appropriate to described these interactions as electrostatic interactions.

We thank the reviewer for identifying this potential inaccuracy. We agree that it may be more appropriate to refer to the interactions as electrostatic and have modified the text as such.

3. L238, please described c-di-GMP as purple stick model.

We have modified the description as suggested.

4. L244, 264, I suggest close up view instead of magnified view in these figure legends.

We have modified the description in the text as suggested.